# Understanding Prejudice and Fidelity of Diverge-to-Converge Multi-Agent Systems

## Abstract

Large language model (LLM) agents have demonstrated substantial potential across various tasks, particularly in multi-agent systems. Among these, *Diverge-to-Converge* (D2C) frameworks stand out for their ability to iteratively diversify and converge intermediate thoughts to improve problem-solving. In this paper, we conduct a comprehensive study on the ***prejudice*** and ***fidelity*** of typical D2C frameworks, including both model-level and society-level frameworks. ❶ In the *prejudice* section, we uncover an inherent *confirmation bias* in D2C systems, which not only leads to suboptimal performance, but also amplifies social biases, such as gender discrimination and political partisanship. Surprisingly, we find that by reframing open-ended problems into controlled initialized problems, this bias can be leveraged to foster more equitable and effective agent interactions, ultimately improving performance. ❷ In the *fidelity* section, we explore the scaling laws of D2C frameworks at different granularities, revealing that increasing the number of agents enhances performance only when the system is not yet saturated—such as in complex tasks or with weaker agents. In saturated scenarios, however, adding more agents can degrade performance. To facilitate further study, we develop `APF-Bench`, a benchmark specifically designed to evaluate such inherent weaknesses of D2C frameworks. We hope our findings offer instructional insights into the strengths and limitations of D2C multi-agent systems, offering guidance for developing more robust and effective collaborative AI systems.

## 1 Introduction

Large language model (LLM) agents have emerged as powerful tools for a wide range of tasks, leveraging their ability to generate human-like text and make decisions across diverse applications (Paranjape et al., 2023; Chen et al., 2021; Tan et al., 2024). Multi-agent systems, in particular, have shown great promise by enabling LLMs to collaborate, compete, and communicate to solve complex problems (Guo et al., 2024a). Among these systems, *Diverge-to-Converge* (D2C) frameworks (Wang et al., 2022; Du et al., 2023; Hong et al., 2023) stand out for their iterative approach: agents independently explore different solutions before converging on an optimized outcome.

However, while D2C frameworks offer significant advantages in enhancing problem-solving capabilities, our work reveals that the iterative nature of these frameworks can introduce inherent challenges that affect both system performance and fairness. In this work, we explore two key challenges: ***prejudice*** and ***fidelity*** in D2C multi-agent systems. We study both *model-level* frameworks, such as `Self-Consistency` (Wang et al., 2022), `Debate` (Liang et al., 2023) and `Consultancy` (Kenton et al., 2024); and *society-level* frameworks, including `CAMEL` (Li et al., 2023), `LangChain` (Pandya & Holia, 2023), and `AutoGen` (Wu et al., 2023).

In the *prejudice* section, we uncover the inherent *confirmation bias* present in D2C systems. Specifically, we find that for various D2C frameworks, the initial stance of the first agent disproportionately influences the outcome. Empirically, we demonstrate that this bias not only leads to suboptimal performance but also amplifies existing social biases in LLMs, such as gender discrimination (Wan et al., 2023) and political partisanship (Rettenberger et al., 2024). Interestingly, we find that reframing open-ended problems as controlled initialized problems by leveraging the potential solution space of a given problem can mitigate this bias, leading to more equitable and effective agent interactions. Notably, the solution space of a given problem can be binary or non-binary as well. For example, in

the debate framework, transforming a math problem into a binary "yes/no" question and initializing the affirmative agent with a wrong answer yields the best performance compared to open-ended debates or correct initialization (see Figure 1).

On the other hand, in the *fidelity* section, we investigate the scaling laws of D2C frameworks by examining the effects of agent count, total token usage, and monetary cost. Our findings show that increasing the number of agents enhances performance only in unsaturated scenarios, such as when tasks are complex or the agents are weaker. In saturated scenarios, however, adding more agents leads to diminishing returns and can even degrade performance due to over-convergence and redundancy.

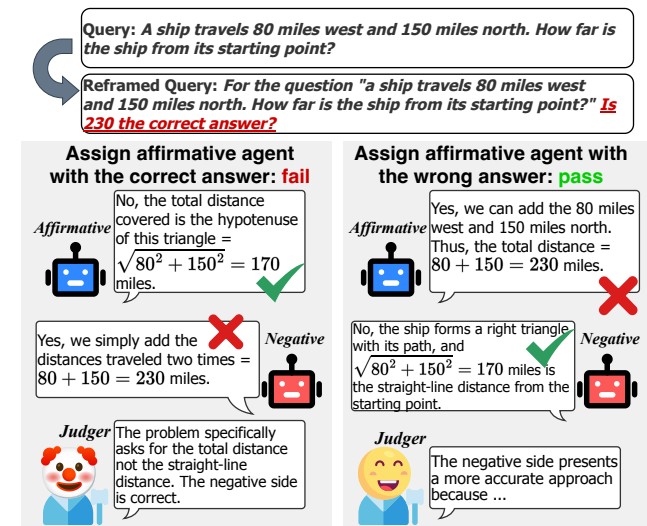

To address these issues, we introduce the **A**gent **P**rejudice and **F**idelity Benchmark (APF-Bench), a benchmark designed to evaluate the prejudice and fidelity inherent in D2C frameworks. Unlike existing benchmarks (Guo et al., 2023b; Zhang et al., 2024; Chen et al., 2024), which focus primarily on task performance, APF-Bench, our benchmark reveals weaknesses related to bias and scalability in agent interactions. Our contributions are summarized as follows:

Figure 1: Example of prejudice in debate framework. In each round, the affirmative and negative agents debate a given question, with a judger summarizing arguments from both sides. By reframing an open-ended math problem into a binary "yes/no" question (top), we find that initializing the affirmative agent with a wrong answer yields better performance compared to open-ended debates or correct initialization (bottom).

- **_Benchmark._** We introduce APF-Bench, a novel benchmark for evaluating the weaknesses in diverge-to-converge (D2C) multi-agent systems, covering both model-level and society-level frameworks. We refine datasets tailored to expose these weaknesses beyond vanilla performance metrics.

- **_Findings._** We systematically analyze how confirmation bias and agent scaling affect D2C frameworks, identifying both performance and fairness issues. Moreover, we have explored the judge bias ratio introduced in multi-agent debates, along with various inherited social biases such as gender bias and political bias.

- **_Remedies._** We propose strategies to mitigate these weaknesses through deliberate system design for various D2C systems, providing insights for optimizing multi-agent LLM systems for more reliable and fairer performance.

## 2 RELATED WORK

**LLM Agents.** LLMs have been widely adopted for *single-agent systems*, where their capabilities in planning, memory, and tool use are leveraged to tackle complex tasks (Guo et al., 2024a; Weng, 2023; Xi et al., 2023). These systems have demonstrated proficiency in decomposing tasks, retaining context, and integrating external tools (Khot et al., 2023; Yao et al., 2023; Shen et al., 2023). However, single-agent systems are limited in complex scenarios requiring collaboration or competition, which has led to the rise of multi-agent systems. Recently, *Multi-agent systems* enable LLMs to interact in complex environments, allowing for collaborative problem-solving and specialization (Guo et al., 2024b). Notably, diverge-to-converge style frameworks (Wang et al., 2022; Liang et al., 2023; Kenton et al., 2024; Li et al., 2023; Hong et al., 2023; Wu et al., 2023), where agents initially explore diverse solutions before converging on a final outcome, have shown promise in applications such as society simulation (Park et al., 2023), software development (Hong et al., 2023), and game simulation (Xu et al., 2024). This work aims to uncover the inferent prejudice and fidelity in multi-agent systems.

**Evaluation of LLM Agents.** Researchers have developed various benchmarks to comprehensively evaluate the efficacy of LLM agents across different tasks. For instance, the RICO dataset (Deka

et al., 2017) is frequently used with Screen2Vec (Li et al., 2021) for evaluating agent performance in mobile interfaces. Benchmarks like the PPTC benchmark (Guo et al., 2023a) assess LLM agents on multi-round dialogue tasks for specific applications, while others such as ToolBench (Qin et al., 2023) and GameBench (Costarelli et al., 2024) evaluate agents' tool use and strategic reasoning abilities, respectively. SWE-Bench (Jimenez et al., 2024) challenges agents to resolve real-world GitHub issues, emphasizing programming proficiency. While these benchmarks are valuable for assessing the general abilities of LLM agents, they often overlook the structural weaknesses inherent in multi-agent systems, particularly those following the diverge-to-converge paradigm. To address this gap, we introduce APF-Bench, a benchmark specifically designed to evaluate the weaknesses inherent in both model-level and society-level multi-agent frameworks.

**Prejudice in LLMs.** LLMs are known to exhibit and amplify biases present in their training data, including societal biases such as gender (Wan et al., 2023; Dong et al., 2024), race (Zhou, 2024; Wang et al., 2024), and political ideology (Rettenberger et al., 2024). These biases can manifest in various applications, from language generation to decision-making, leading to unfair or harmful outcomes. Prior work has explored the ways in which LLMs reinforce stereotypes (Kotek et al., 2023). Approaches like adversarial training (Ernst et al., 2023), data augmentation (Raza et al., 2024), and prompt engineering (Kamruzzaman & Kim, 2024) have been proposed to mitigate these biases. Our work builds on this by focusing on how bias influences multi-agent interactions within diverge-to-converge frameworks, providing insights into the amplification and mitigation of biases.

## 3 PRELIMINARIES: DIVERGE-TO-CONVERGE FRAMEWORKS

In this section, we formally define the two levels of multi-agent frameworks evaluated in APF-Bench: model-level and society-level multi-agent systems. Diverge-to-converge (D2C) frameworks are characterized by an iterative process where agents initially diverge by exploring different potential solutions and later converge by synthesizing these diverse outputs into a final solution. These frameworks rely on agents either collaborating or competing to improve decision-making through this cyclical divergence and convergence. To quantify this process, we introduce a parameter $C$, which tracks the total number of calls made to any LLM during task execution. Illustrative examples of these frameworks are provided in Figure 2.

### 3.1 MODEL-LEVEL MULTI-AGENT SYSTEM

In model-level frameworks different agents operate at the model level to collaborate or compete toward a consensus or solution. We describe three specific D2C model-level frameworks: self-consistency, debate, and consultancy, with $C$ denoting the total number of LLM calls.

▷ Self-Consistency. In self-consistency, an agent $A$ generates multiple independent solutions $\{s_1, s_2, \ldots, s_n\}$ for a given problem $P$. These solutions are then aggregated, typically by majority voting or averaging, to produce a final consistent output $s^*$. Divergence occurs when the agent explores different possible solutions, while convergence happens when the system aggregates them. The total number of LLM calls is $C = n$, where $n$ solutions are generated and processed. Formally, this is defined as: $s^* = \text{Aggregate}(\{A(P, \theta_1), A(P, \theta_2), \ldots, A(P, \theta_n)\})$, where $\theta_i$ represents the different configurations of the agent.

▷ Debate. In the debate framework, two agents $A_{\text{aff}}$ (Affirmative) and $A_{\text{neg}}$ (Negative) argue over the correctness of their respective solutions to problem $P$ in up to $t$ rounds. Divergence occurs as both agents present different arguments in each round, and convergence is achieved when a judger $J$ (another LLM) decides on the final outcome. The judger can terminate the debate early if a conclusive decision is reached before round $t$. The total number of LLM calls is $C = 3t'$, where $t' \leq t$ represents the number of completed rounds. Mathematically, the debate is expressed as:

$$\text{Debate}(A_{\text{aff}}, A_{\text{neg}}, J, P, t') = (A_{\text{aff}}^{(t')}(P), A_{\text{neg}}^{(t')}(P), J^{(t')}(A_{\text{aff}}^{(t')}(P), A_{\text{neg}}^{(t')}(P))),$$

where the judger's decision at round $t'$ is: $s^* = J^{(t')}(A_{\text{aff}}^{(t')}(P), A_{\text{neg}}^{(t')}(P))$.

▷ Consultancy. The consultancy framework consists of two agents: a primary decision-maker $A_{\text{primary}}$ and a consultant $A_{\text{consult}}$. Divergence happens as the consultant offers iterative feedback over multiple rounds, helping the primary agent refine its solution. Convergence occurs when the primary

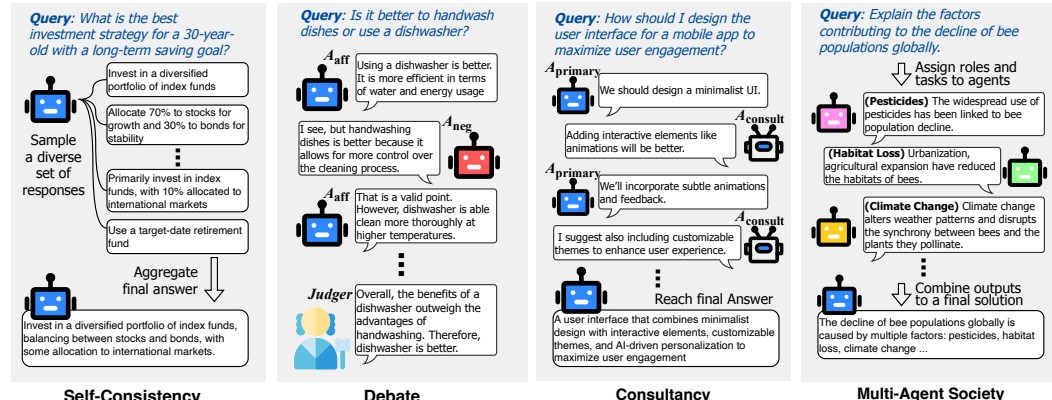

Figure 2: Overview of model-level and society-level multi-agent frameworks.

agent finalizes its solution based on the consultant's feedback. The primary agent can terminate the process early if a satisfactory solution is reached. The total number of LLM calls is $C = 2t'$, where $t' \leq t$ represents the completed rounds. Formally, consultancy is expressed as:

$$\text{Consultancy}(A_{\text{primary}}, A_{\text{consult}}, P, t') = (A_{\text{primary}}^{(t')}(P), A_{\text{consult}}^{(t')}(A_{\text{primary}}^{(t')}(P))),$$

with the final solution: $\quad s^* = A_{\text{primary}}^{(t')}(P, A_{\text{consult}}^{(t')}(A_{\text{primary}}^{(t')}(P)))$.

## 3.2 SOCIETY-LEVEL MULTI-AGENT SYSTEM

In society-level D2C frameworks, multiple autonomous agents operate independently but collaborate to solve a larger task. Divergence occurs as each agent is assigned a specific role $r_i$ and tackles a different sub-task, and convergence happens when their outputs are combined into a final solution. Each agent is responsible for a specific sub-task $P_i$ of the overall task $P$, and the total number of LLM calls is given by $C = \sum_{i=1}^{m} C_i$, where $C_i$ is the number of calls made by agent $A_{r_i}$. Formally, the process is defined as: $\quad S(P) = \text{Combine}(\{A_{r_1}(P_1), A_{r_2}(P_2), \ldots, A_{r_m}(P_m)\})$, where Combine is the strategy for integrating sub-task solutions into the overall task solution.

Examples of society-level frameworks include AutoGen (Wu et al., 2023), CAMEL (Li et al., 2023), and LangChain (Pandya & Holia, 2023). These systems emphasize different aspects of agent collaboration. AutoGen allows for dynamic agent interactions across diverse domains, while CAMEL focuses on role-based cooperation with minimal human guidance. LangChain structures complex task dependencies through graph-based modeling. In all these systems, the efficiency of collaboration is crucially impacted by the total number of LLM calls, which directly affects computational cost.

## 4 DEFINITIONS OF PREJUDICE AND FIDELITY

In this section, we formally define the key concepts of *Prejudice* and *Fidelity* as they apply to Diverge-to-Converge (D2C) multi-agent systems. These definitions allow us to systematically analyze the effects of agent interactions and scaling behavior within D2C frameworks.

### 4.1 PREJUDICE IN D2C FRAMEWORKS

**Definition 1.** ***Prejudice*** *in D2C frameworks refers to the inherent biases introduced by the initial conditions of the system, which disproportionately affect the final outcomes.*

This concept of prejudice aligns with the notion of *confirmation bias*, where the initial stance or role of an agent skews the final decision-making process.

Specifically, in D2C frameworks, prejudice menifest as follows: the initial stance of the first agent proposer significantly influences the final outcome. For instance, in a debate framework, the initial argument presented by the affirmative agent can disproportionately sway the debate's conclusion,

especially if subsequent rounds reinforce the initial stance rather than challenge it. Formally, let $A_{\text{init}}$ be the initial agent's stance, and $s^*$ be the final solution. The influence of $A_{\text{init}}$ on $s^*$ is defined as:

$$\text{Prejudice} = \frac{\partial s^*}{\partial A_{\text{init}}}$$

A high value of Prejudice indicates that the initial agent's stance heavily influences the final outcome, suggesting confirmation bias.

In practice, we use the performance variation under changed conditions to indicate the prejudice. Prejudice not only affects the performance of D2C systems but also amplifies underlying social biases embedded in large language models (LLMs), such as gender discrimination (Wan et al., 2023) and political partisanship (Rettenberger et al., 2024). These biases, if unchecked, can propagate through agent interactions, leading to unfair or skewed outcomes.

## 4.2 Fidelity in D2C Frameworks

**Definition 2.** *Fidelity in D2C frameworks refers to the system's ability to enhance performance as the number of agents, computational resources, or data scales.*

Fidelity examines how well the system's architecture supports effective collaboration and decision-making without performance degradation. It encompasses four main dimensions:

**1. Agent Count.** The number of agents, which is characterized by the number of LLM calls $C$, in a D2C framework plays a critical role in performance. Let $P(C)$ represent the system's performance. Fidelity is depicted by how $P(C)$ scales as $C$ increases.

**2. Total Token Usage.** The total number of tokens processed by the LLMs during multi-agent interactions impacts the computational cost. Let $T_{\text{total}}$ denote the total tokens used by all agents:

$$T_{\text{total}} = \sum_{i=1}^{C} T_i$$

where $T_i$ is the number of tokens processed by agent $A_i$. High fidelity in this dimension implies that the system can handle increased token usage without incurring significant computational overhead or loss in performance.

**3. Monetary Cost.** The financial cost of running D2C systems scales with the number of tokens processed and the number of agents involved. Let $M_{\text{monetary}}$ represent the monetary cost, which is proportional to $T_{\text{total}}$ and the number of agents: $M_{\text{monetary}} = m \cdot T_{\text{total}}$, where $m$ is the cost per token. High fidelity in this dimension means that the system maintains performance at a reasonable cost as resources scale.

By defining prejudice and fidelity in this formal manner, we provide a framework for evaluating both the inherent biases and the scaling behavior of D2C multi-agent systems. These metrics help identify the conditions under which D2C frameworks succeed or fail, offering insights for optimizing performance and fairness.

## 5 Evaluation Strategy for Prejudice & Fidelity

**Evaluation Datasets.** We employ several datasets that evaluate a broad range of reasoning and problem-solving abilities under the impact of prejudice and fidelity. Each dataset focuses on a specific task type, and we use unique metrics associated with these datasets to evaluate the models comprehensively. We propose to refine the datasets by reducing the trivial samples as detailed in Section 5. We list the sample numbers for the datasets here.

- *PIQA* (Bisk et al., 2020): A dataset for two-choice physical interaction question answering, designed to test commonsense physical reasoning *i.e.,* given a question either "statement-1" or "statement-2" is correct which essentially forms a binary solution space. The metric used is *Accuracy*, measuring how often the model selects the correct answer. The original dataset contains 1837 samples, and the refined version has 304 samples.

- *StrategyQA* (Geva et al., 2021): A dataset for binary yes or no questions that requires implicit reasoning. The evaluation metric is *Yes/No Accuracy*, which assesses the proportion of correct answers based on implicit strategies derived by the model. The original dataset contains 2290 samples, and the refined version has 297 samples.
- *GSM8K* (Cobbe et al., 2021): A dataset for grade-school-level arithmetic and mathematical reasoning, where models solve math word problems with a non-binary solution space *i.e.* lies in $\mathbb{R}$ (real numbers). The metric is *Accuracy*, evaluating the correctness of the final answer. The original dataset contains 1319 samples, and the refined version has 300 samples.
- *Chess Move Validity* (Srivastava et al., 2022): This dataset assesses strategic reasoning through chess move predictions, with performance measured by the *Pawn Score* (evaluated by Stockfish), indicating the quality of predicted moves. This dataset has a non-binary solution space as there exist 64 possible answers formatted as $[a - h][1 - 8]$ representing potential valid and non-valid chess moves. The actual number of valid moves may vary depending on the specific state of the chessboard (e.g., piece positions, legal moves). Each generated answer was deemed correct as long as it was one of the valid answers in the sequence. The original dataset contains 1000 samples, and we did not apply refinement since the complexity of the dataset is already high.
- *Debatepedia* (Kobbe, 2019): A dataset including real-world controversial debating questions online. This dataset has no task labels, and we utilize this dataset to measure the social bias of D2C frameworks and their induced scaling law. The original dataset contains 573 samples, and we did not apply refinement since the complexity of the dataset is already high.

These datasets cover diverse reasoning tasks and enable a comprehensive evaluation of the models across domains like commonsense reasoning, debate, factual reasoning, and strategy.

**Dataset Problem Reframing.** To explicitly evaluate the confirmation bias, we propose a problem reframing technique to transform open-ended questions into controlled initialized questions by leveraging the potential solution space of a given problem. This reframing approach allows us to explicitly evaluate the confirmation bias on how initial conditions of the D2C systems affect the final outcomes. For example, the original open-ended questions require models to generate specific answers,

> *What is the answer to 3 + 4?*

We transform this into a controlled initialized question format, which asks whether a proposed value $\{x\}$ is the correct answer to the problem:

> *For the question "What is the answer to 3 + 4", is $\{x\}$ the correct answer?*

Here, $x$ can either be the ground-truth answer (in this case, 7) or an incorrect answer (*e.g.*, 9). We refer to this as the **control reframing**. We further distinguish between two conditions of control reframing as follows:

- *Control (Right) Reframing*: This occurs when $x$ is the ground-truth answer. In our example, $x = 7$, the question asks if 7 is the correct answer to 3 + 4. This framing allows the model to directly confirm whether the correct answer is valid.
- *Control (Wrong) Reframing*: This occurs when $x$ is an incorrect answer. For instance, if $x = 9$, the model is asked whether 9 is the correct answer to 3 + 4. This framing introduces a deliberate mismatch, testing the model's ability to identify incorrect answers and adjust accordingly.

By transforming open-ended problems into controlled initialized questions, we gain a more controlled environment to assess how models respond under different conditions. This reframing technique enables us to observe how D2C systems handle biases in problem formulation and whether they can maintain robust performance when faced with diverse options, as in the controlled cases. In Appendix A, we detail the proposed controlled initialization framework for the datasets.

**Dataset Refinement.** The original datasets used in our evaluation contain many simple questions that result in high accuracy across all models, making it difficult to distinguish the performance of different models and frameworks. To address this issue, we refine the datasets by selecting samples where not all models make correct predictions, focusing on more challenging examples that are likely to expose meaningful performance differences between multi-agent systems. The dataset refinement process follows these steps:

- *Correctness Evaluation:* We first evaluate each model on every sample in the dataset. For each sample, we count how many models make the correct prediction. Samples where all models provide correct predictions are discarded.
- *Framework-Control Sampling:* From the pool of framework-control combinations (e.g., `Debate` with control wrong), we randomly select three combinations. These combinations are applied to the refined dataset to further evaluate model performance under different conditions.
- *Final Sampling:* After applying the framework-control settings, we sort the refined dataset by the number of correct answers (i.e., from the hardest samples, where the fewest models were correct, to the easiest). We then select the top 300 samples with the fewest correct answers to ensure we focus on the most challenging and informative cases.

A pseudo-code style descrpition in given in Algorithm 1 below.

---

**Algorithm 1** Dataset Refinement for Multi-Agent Frameworks

---

**Input:** Dataset $D$, Models $M = \{Model_1, Model_2, \ldots, Model_k\}$
**Output:** Refined Dataset $D_{\text{refined}}$

**Step 1:** Initialize an empty set $D_{\text{refined}}$ with tuples $(s, correct\_prediction\_count)$
**Step 2: for** sample $s$ in Dataset $D$ **do**
    Initialize $correct\_prediction\_count = 0$ **for** model $m$ in Models $M$ **do**
        Evaluate model $m$ on sample $s$ **if** model $m$'s prediction for $s$ is correct **then**
            Increment $correct\_prediction\_count$
        **end**
    **end**
    **if** $correct\_prediction\_count < |M|$ **then**
        Add tuple $(s, correct\_prediction\_count)$ to $D_{\text{refined}}$
    **end**
**end**
**Step 3:** Randomly select 3 (framework, control) combinations from pool $P$ Let $Selected\_P = \{(Framework_i, Control_i), (Framework_j, Control_j), (Framework_k, Control_k)\}$
**Step 4: for** (framework, control) combination in $Selected\_P$ **do**
    **for** sample $s$ in $D_{\text{refined}}$ **do**
        Apply the framework with the control setting to sample $s$ (e.g., Debate with control wrong) Evaluate model performance under this control setting Record the results for analysis
    **end**
**end**
**Step 5:** Sort $D_{\text{refined}}$ in ascending order by $correct\_prediction\_count$
**Step 6:** Select the top 300 samples with the fewest correct answers $D_{\text{final}}$ = top 300 samples from $D_{\text{refined}}$
**Step 7:** Return $D_{\text{final}}$

---

# 6 APF-Bench: THE AGENT PREJUDICE AND FIDELITY BENCHMARK

In this section, we evaluate the performance of various multi-agent frameworks using our proposed evaluation metrics, focusing on both task performance and the impact of prejudice and fidelity. We conduct a series of experiments on the refined datasets to compare the effectiveness of different D2C multi-agent frameworks across multiple scenarios. Details of the frameworks are given in Section 3. We initialize LLM agents employing GPT-4o, GPT-4o-mini, Gemini-Pro, and Gemini-Flash with identifiers `gpt-4o-2024-05-13`, `gpt-4o-mini-2024-07-18`, `gpt-4o-mini-2024-07-18`, `gemini-1.5-flash-001`, and `gemini-1.5-pro-001` respectively. Without further claims, their default temperatures are to 0 to increase the stability. Further implementation details are presented in Appendix B. All the used system prompts with reference are given in Appendix H.

## 6.1 PREJUDICE

**Setup:** To evaluate prejudice across different frameworks, we follow the default hyper-parameter guidelines outlined in the original papers. For the `Self-Consistency` framework, we set the number of LLM agents to $n = 4$. In the `Debate` and `Consultancy` frameworks, we set the maximum number of rounds to $t = 5$. For experiments involving the Debatepedia dataset, we extend the round limit to $t = 10$ due to the controversial topics requiring more extensive communication to explore differing perspectives and reach a resolution. For society-level frameworks, including `Camel`, `Langchain`, and `AutoGen`, where interactions among agents are more frequent, we set slightly smaller values for $n$ and $t$ for fair comparisons. Specifically, we configure the number of agents as $n = 3$ and the maximum

Table 1: Performance of different model-level D2C frameworks under different controlled settings. For brevity, we use open for open-ended, **CR** for controlled (right), and **CW** for controlled (wrong). Metrics for the datasets are listed in Section 5, with reported scores in %. The highest score for each control condition comparison is **bold**. See Table 2 in Appendix D for results of society-level frameworks.

| | Self-consistency | | | Debate | | | Consultancy | | |
|---|---|---|---|---|---|---|---|---|---|
| | **Open** | **CR** | **CW** | **Open** | **CR** | **CW** | **Open** | **CR** | **CW** |
| *PIQA* | | | | | | | | | |
| GPT-4o | 93.01% | 94.23% | **94.71**% | 92.00% | 93.32% | **94.71**% | 91.73% | 93.01% | **93.32**% |
| GPT-4o-mini | 78.26% | 80.34% | **82.73**% | 79.38% | 82.22% | **85.09**% | 78.75% | 81.74% | **83.51**% |
| Gemini-Pro | 84.31% | 86.73% | **88.34**% | 82.00% | 85.77% | **89.24**% | 83.51% | 85.09% | **88.34**% |
| Gemini-Flash | 80.34% | 83.51% | **84.31**% | 78.74% | 82.73% | **84.73**% | 79.38% | 82.22% | **84.31**% |
| *StrategyQA* | | | | | | | | | |
| GPT-4o | 79.67% | 78.33% | **79.67**% | **80.33**% | 79.33% | 79.67% | 79.67 % | 79.33% | **79.67**% |
| GPT-4o-mini | 70.33% | 71.67% | **72.00**% | 69.67% | 72.33% | **73.00**% | 69.67% | 71.67% | **72.33**% |
| Gemini-Pro | 76.33% | **77.33**% | 76.00% | 74.33% | 75.67% | **77.00**% | 73.67% | 75.00% | **75.67**% |
| Gemini-Flash | 70.67% | 71.67% | **72.33**% | 68.67% | 71.67% | **72.67**% | 70.33% | 71.33% | **72.00**% |
| *GSM8K* | | | | | | | | | |
| GPT-4o | 91.67% | 93.00% | **93.67**% | 89.67% | 93.00% | **93.33**% | 90.00% | 91.67% | **91.67**% |
| GPT-4o-mini | 86.67% | 91.33% | **92.33**% | 87.67% | 90.00% | **92.33**% | 86.67% | 90.33% | **92.00**% |
| Gemini-Pro | 91.33% | 92.67% | **93.33**% | 89.00% | 91.67% | **94.00**% | 91.00% | 92.33% | **93.33**% |
| Gemini-Flash | 86.33% | 90.67% | **92.00**% | 86.67% | 91.67% | **92.33**% | 87.67% | 91.33% | **92.67**% |
| *Chess Move Validity* | | | | | | | | | |
| GPT-4o | 70.70% | 72.20% | **74.70**% | 66.70% | 73.90% | **76.40**% | 69.30% | 71.90% | **73.70**% |
| GPT-4o-mini | 35.30% | 43.80% | **48.70**% | 44.90% | 48.70 % | **51.10**% | 38.20% | 42.60% | **45.90**% |
| Gemini-Pro | 47.60% | 50.40% | **52.70**% | 43.80% | 50.60% | **53.70**% | 45.60 % | 49.80 % | **53.00**% |
| Gemini-Flash | 38.20% | 39.70% | **42.50**% | 31.20% | 41.20% | **44.50**% | 33.40% | 40.30% | **42.80**% |

number of rounds to $t = 4$. **Note:** In this subsection, our primary goal is to explore the prejudice in these frameworks as they approach their upper-bound performance. Therefore, the number of API calls is chosen based on when we observe approximately saturated performance for each framework. A more comprehensive study on the scaling behaviors of different models is provided in Section 6.2.

### 6.1.1 CONFIRMATION BIAS

We evaluate confirmation bias by performing condition-control evaluations, fixing the initial stance of the first agent, and measuring its influence on the outcome. Table 1 presents the results for model-level frameworks. Results for society-level frameworks are given in Table 2 in Appendix D due to space limitation. Our key findings are as follows.

▷ **LLM and Framework Variability:** Different LLMs excel at different tasks. For example, Gemini models perform better in mathematical reasoning (e.g., Gemini-Pro achieves **94%** accuracy on GSM8K), while GPT models outperform in chess (e.g., GPT-4o achieves **76.4%** *pawn* advantage). Larger models, such as GPT-4o and Gemini-Pro, consistently perform better than their smaller versions (e.g., GPT-4o-mini and Gemini-Flash).

▷ **Controlled Initialization via Problem Reframing:** One of the key insights from the experiments is that D2C frameworks using the proposed controlled initialization through problem reframing consistently outperform those using open-ended approaches. The performance difference is particularly evident for less capable or smaller LLMs, such as GPT-4o-mini and Gemini-Flash, whose performances are less saturated. For example, in the Chess Move Validity dataset, GPT-4o-mini shows a **6.2%** improvement with control (right) reframing compared to open-ended initialization. This improvement is more modest for larger models like GPT-4o, where the performance gains are smaller but still present. The results suggest that the controlled initialization method helps smaller models converge to better solutions by narrowing their search space.

▷ **Surprising Effectiveness of Control (Wrong) Reframing:** We speculate that this occurs because D2C frameworks encourage exploration and favor an outcome that diverges from the initial stance. This shows that such confirmation bias can be leveraged to promote model performance. This is particularly evident in frameworks like Debate and Consultancy, where agents tend to adjust their final decisions away from the initial proposition. For example, in the Chess-Moving dataset, Control (Wrong) Reframing leads to a **3.3%** improvement over Control (Right) Reframing for smaller models like Gemini-Flash. This finding is beneficial for practical applications, as reframing an open-ended task into a controlled question with a random wrong answer can lead to near-optimal performance, approaching the results seen in Control (Wrong) Reframing.

▷ **Control (Right) Reframing Outperforms Open-Ended:** This highlights the first level of confirmation bias, where presenting the ground-truth answer as a controlled right initialized question helps narrow the search space for LLM agents. For example, on the PIQA dataset, Control (Right) Reframing leads to a **5.71%** improvement over the open-ended approach for GPT-4o-mini. This shows that framing a task in a way that anchors agents to the correct answer early in the process helps boost performance, particularly for smaller and less capable models.

### 6.1.2 FURTHER INVESTIGATION

▷ **Affirmative vs. Negative Agent Bias.** Figure 3 compares the bias ratio of judges towards affirmative versus negative agents in both Open Debate (*left*) and Control Debate (*middle*: control right, *right*: control wrong) scenarios across three datasets (GSM8k, PIQA, StrategyQA). In the Open Debate setting, affirmative agents tend to have slightly less bias compared to negative agents. This trend is consistent across datasets. In the Control Debate setting, however, the affirmative bias decreases significantly, particularly for PIQA and StrategyQA, while the negative bias increases. The shift in bias ratios between Open and Control debates indicates that control mechanisms (rules or restrictions) during the debate substantially influence the way judgments are made. In the Open Debate, the balance between affirmative and negative agents seems evener, while in the Control Debate, the imposed structure might skew bias toward negative agents due to stricter interpretation or framing of arguments. This suggests that the controlled initialization amplifies the confirmation biases and is utilized to enhance performance.

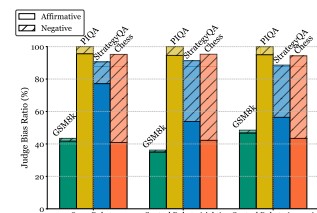

Figure 3: Illustration of the judge bias ratio towards affirmative and negative agents in debate.

▷ **Social Bias.** Figure 4 shows the influence of social biases in GPT-4, specifically focusing on bias in confirmation (*left*), ideology (*middle*), and sex (*right*), in the `Debatedpedia` dataset. The bars show a significant bias in favor of affirmative agents for confirmation bias, ideology (left-leaning), and sex (female). Negative bias in these areas is noticeably lower. The model exhibits a clear tendency towards affirmative agents for confirmation bias, which suggests that the model is more inclined to agree or favor responses that confirm pre-existing beliefs or assertions. Regarding ideological bias, the model displays a left-leaning inclination, favoring left-ideology over right-ideology, which could raise concerns regarding political neutrality. The sex bias shows a preference for female agents over male agents, highlighting gender bias within the model's responses, which may need to be addressed to ensure fairness and equitable representation.

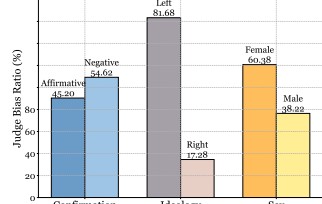

Figure 4: Illustration of the judge bias ratio towards affirmative or negative (*right*), left or right (*middle*), and female or male (*left*) agents in debate.

## 6.2 FIDELITY

In this subsection, we examine the fidelity of society-level D2C frameworks, which is an essential component in evaluating the performance of multi-agent systems when computation scales. We focus on four key aspects that influence fidelity: the number of API calls, the number of tokens, and the financial cost observed across different datasets. These factors are closely tied to the system's capacity to maintain high fidelity while scaling, and each presents unique trade-offs in performance.

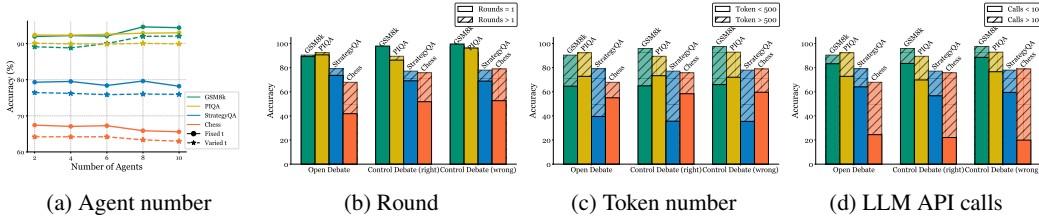

| (a) Agent number | (b) Round | (c) Token number | (d) LLM API calls |

Figure 5: The averaged performance v.s. resources of `GPT-4o` in multi-agent frameworks on four datasets. with different (a) agent numbers, (b) debate rounds, (c) token numbers, and (d) LLM API calls.

▷ **Scaling Law is Effective:** Our results show that increasing the number of LLM calls, tokens, or agents generally leads to better performance, particularly in complex tasks. For example, as shown in Figure 6 (b) and detailed statistics in Appendix F, in tasks such as Chess Move Validity, having more debate rounds or adding more agents significantly improves strategic diversity and leads to better

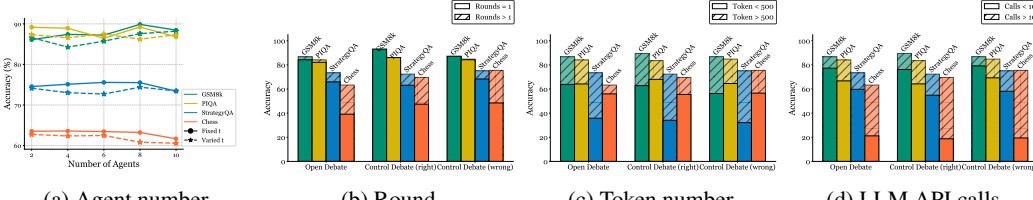

|(a) Agent number|(b) Round|(c) Token number|(d) LLM API calls|

Figure 6: The averaged performance v.s. resources of `GPT-4o-mini` in society-level frameworks on four datasets with different (a) agent numbers, (b) debate rounds, (c) token numbers, and (d) LLM API calls.

intermediate outputs. Similarly, in tasks involving weaker models, like GPT-4o-mini, performance increases as more agents are introduced, supporting the idea that scaling these values can improve outcomes. In society-level frameworks, where agents interact with predefined roles, scaling has an even more pronounced effect due to the added layers of communication and collaboration. Moreover, when the temperature of LLMs is varied, the improvements brought by the scaling law are still consistent as presented in Figures 5 (a) and 6 (a), which indicates the robustness of the scaling law against variance during inference.

▷ **Saturation Occurs in Easier Tasks:** While scaling improves performance in more complex tasks, we observed that saturation occurs in easier tasks where increasing the number of agents or tokens no longer contributes to performance gains as shown in Figures 5 and 6 (a) and (c). For instance, in PIQA dataset, performance saturates when adding more agents beyond a certain point of 4 agents. In these cases, larger models like GPT-4o and Gemini-Pro also experience diminishing returns, showing that once a task becomes trivial for the framework, additional scaling no longer helps. This effect is especially noticeable in model-level frameworks, where simpler tasks quickly reach saturation point.

▷ **Too Many Rounds Can Decrease Performance, But Not Tokens:** One notable finding is that adding too many rounds of interactions between agents can negatively impact performance, especially in society-level frameworks, due to increased coordination complexity. This is particularly evident in Figures 5 (b) and 6 (b) for tasks like GSM8K, where most correct answers are achieved with only 1 round. This is because agents may struggle with redundant processing and conflicting outputs when the number of rounds is large. However, other metrics like token usage do not show the same performance decrease. For example, in tasks like StrategyQA, increasing the token count improves the quality of responses without leading to performance drops, as long as the task complexity justifies the additional information exchange.

▷ **More LLM Calls Are Helpful in Complex Tasks:** A higher number of LLM calls is especially beneficial for complex reasoning tasks as shown in Figures 5 (d) and 6 (d), such as those found in the Chess Move Validity dataset. Generally, more LLM calls allow for better problem-solving through diverse strategies and collaborative refinement. For instance, in society-level frameworks, agents with predefined roles collaborate more effectively, leading to substantial problem-solving improvements.

▷ **Monetary Cost.** Figure 7 compares the average computational costs (in dollars) of model-level and society-level frameworks using GPT-4o and GPT-4o-mini across four datasets Society-level frameworks incur significantly higher costs than model-level frameworks

Figure 7: The average costs for model-level and society-level frameworks using GPT-4o and GPT-4o-mini on four datasets.

across all datasets, particularly for complex tasks like Chess and GSM8k. Chess, in particular, shows the highest cost in society-level frameworks, likely due to the complexity and coordination required among multiple agents. The costs across all datasets reflect the relative complexity of the tasks, with Chess demanding the most resources, followed by GSM8k.

## 7 CONCLUSION

We investigate the challenges of prejudice and fidelity in Diverge-to-Converge (D2C) multi-agent frameworks. Our findings revealed inherent confirmation bias that affects performance and amplifies social biases. We demonstrated that reframing open-ended problems into controlled (right or wrong) initialized questions can utilize these biases to enhance performance. Furthermore, we explored the scaling laws of D2C systems, showing that increasing agent numbers improves performance only in unsaturated scenarios, while saturated systems suffer from over-convergence and redundancy.

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

CONTENTS

APPENDIX

## A CONTROLLED INITIALIZATION

Here, we present controlled initialization through problem reframing for multi-agent systems by leveraging the potential solution space of a given problem.

Let, $Q$ denote a question, $S_R$ the correct answer space, and $S_W$ the wrong answer space. Controlled initialization is structured as follows:

- For controlled wrong initialization, the prompt is: *"Is $S_W$ the correct answer to the question?"*
- For controlled right initialization, the prompt is: *"Is $S_R$ the correct answer to the question?"*

As discussed earlier, PIQA and StrategyQA have a binary solution space whereas GSM8K and Chess Move Validity datasets exhibit a non-binary solution space. So for binary tasks like PIQA and StrategyQA, where $S_W =\sim S_R$, the solution space is straightforward. However, for non-binary tasks:

- GSM8K: $S_R \subset \mathbb{R}$ (correct numerical solution), and $S_W = \mathbb{R} \setminus S_R$.
- Chess Move Validity: $S_R$ is the set of valid answers out of 64 possible moves, and $S_W$ encompasses the complement of valid moves.

***Broader Applicability and Frameworks:*** These controlled initializations are initiated on the affirmative side during the starting round for both Multi-label frameworks (e.g., Self-consistency, Debate, Consultancy) and Society-label frameworks (e.g., CAMEL, LangChain, AutoGen). By explicitly tailoring the initialization to the solution space, we maintain flexibility to address task-specific complexity.

## B IMPLEMENTATION DETAILS

Our implementation is available at https://github.com/apf-bench/APF-Bench. For our experiments, we employed a range of large language model (LLM) agents, initializing them with various versions and identifiers to assess the Diverge-to-Converge (D2C) frameworks. The LLM agents used were:

- **GPT-4o**: Identifier `gpt-4o-2024-05-13`
- **GPT-4o-mini**: Identifier `gpt-4o-mini-2024-07-18`
- **Gemini-Pro**: Identifier `gemini-1.5-pro-001`
- **Gemini-Flash**: Identifier `gemini-1.5-flash-001`

These models were selected for their robust handling of complex multi-agent scenarios and their scalability in both small and large agent settings. Each model was initialized with the default temperature set to 0 to maximize the stability of responses and minimize randomness, ensuring consistent evaluation during multi-agent coordination and convergence processes.

### B.1 MODEL CONFIGURATION AND SYSTEM PROMPTS

The system prompts and configurations used for initializing the agents followed best practices for enhancing agent cooperation and alignment with task objectives. The detailed system prompts, including task-specific instructions, can be found in Appendix H. These prompts were adapted to fit the nuances of each model's strengths and were designed to reduce bias amplification and improve fidelity during agent interactions.

### B.2 HARDWARE AND COMPUTATIONAL SETUP

The experiments were conducted on CPUs only with dependecies on the access to the LLM API from OpenAI and Google, using distributed computing to handle the scale of multi-agent interactions. We optimized the memory and compute requirements by dynamically adjusting the number of agents and tasks across different frameworks, ensuring that performance remained consistent without overloading computational resources.

## C  IMPACT FROM THE TEMPERATURE OF LLMs

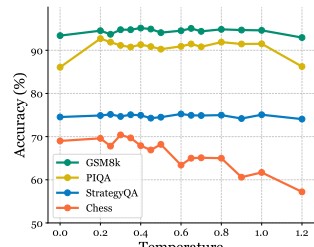

Figure 8 presents the influence of different temperature settings (ranging from 0 to 1.2) on the accuracy of GPT-4 across the same datasets (GSM8k, PIQA, StrategyQA, and Chess). The accuracy for most datasets (GSM8k, PIQA, StrategyQA) appears relatively stable across varying temperature values. However, Chess demonstrates a notable drop in performance as the temperature increases. Temperature in the GPT model controls randomness in response generation, and as temperature increases, responses become more diverse. The stable performance at various temperature values for most datasets suggests that those tasks are less sensitive to diversity and can maintain high accuracy regardless of randomness. The Chess dataset's performance drop indicates that for strategic tasks,

Figure 8: The influence of different temperatures for a n agents in `Self-Consistency`.

higher randomness might confuse the model, leading to less accurate answers. Lower temperature values (less randomness) might be required to ensure consistency in these types of task.

## D  PERFORMANCE OF SOCIETY-LEVEL D2C FRAMEWORKS

Similar to Table 1 in the main paper, Table 2 below reports the performance of three widely adopted society-label D2C frameworks.

Table 2: Performance of different society-level D2C frameworks under different controlled settings. For brevity, we use open for open-ended, **CR** for controlled (right), and **CW** for controlled (wrong). Metrics for the datasets are listed in Section 5, with reported scores in %. The highest score for each control condition comparison is **bold**.

| | Langchain | | | CAMEL | | | AutoGen | | |
| --- | --- | --- | --- | --- | --- | --- | --- | --- | --- |
| | Open | CR | CW | Open | CR | CW | Open | CR | CW |
| *PIQA* | | | | | | | | | |
| GPT-4o | **90.35%** | 94.23% | **94.71%** | 89.80% | **94.78%** | 91.98% | 94.31% | **92.83%** | 92.55% |
| GPT-4o-mini | 81.17% | 84.36% | 81.63% | 82.43% | 78.52% | **81.35%** | 79.46% | 83.83% | **86.89%** |
| Gemini-Pro | 85.53% | **89.28%** | 86.93% | 84.12% | 88.67% | **85.57%** | 84.49% | 89.72% | **90.14%** |
| Gemini-Flash | 79.76% | 81.80% | **83.64%** | 79.14% | 82.15% | **82.94%** | **81.35%** | 85.63% | 82.31% |
| *StrategyQA* | | | | | | | | | |
| GPT-4o | **79.22%** | 82.58% | 79.93% | **80.33%** | 79.33% | 79.67% | 81.26% | 77.51% | **80.18%** |
| GPT-4o-mini | 74.35% | 70.22% | 68.93% | 71.68% | 70.39% | **74.58%** | 67.91% | 72.08% | **71.64%** |
| Gemini-Pro | 75.83% | **79.53%** | 73.13% | 72.71% | 76.37% | **75.33%** | 70.48% | 77.29% | **79.35%** |
| Gemini-Flash | 74.44% | 74.48% | **74.69%** | 66.24% | 73.22% | **74.13%** | 73.32% | 72.64% | **73.41%** |
| *GSM8K* | | | | | | | | | |
| GPT-4o | 88.99% | **94.43%** | 93.22% | 92.18% | 93.73% | **92.67%** | 91.24% | 93.24% | 91.02% |
| GPT-4o-mini | 85.12% | 88.46% | **92.12%** | 83.34% | 87.49% | **93.51%** | 85.95% | 88.92% | **92.64%** |
| Gemini-Pro | 88.53% | 91.99% | **94.18%** | 85.97% | 93.16% | **91.58%** | 89.27% | 92.71% | **94.13%** |
| Gemini-Flash | 88.54% | 87.69% | **88.61%** | 85.13% | 90.65% | **94.12%** | 90.46% | 91.95% | **92.48%** |
| *Chess Move Validity* | | | | | | | | | |
| GPT-4o | 67.40% | 71.30% | **75.60%** | 68.60% | 72.30% | **76.20%** | 66.80% | 72.40% | **73.10%** |
| GPT-4o-mini | 37.10% | 45.60% | **48.10%** | 42.70% | 47.90% | **49.40%** | 39.10% | 40.90% | **46.70%** |
| Gemini-Pro | 45.70% | 49.10% | **54.10%** | 46.40% | 50.40% | **52.80%** | 46.20% | 50.20% | **54.90%** |
| Gemini-Flash | 39.60% | 40.80% | **43.20%** | 32.90% | 43.20% | **46.00%** | 34.70% | 42.80% | **44.70%** |

## E  FURTHER DISCUSSION

*Comparison with Other LLM Debias Directions.* The biases studied in this work are identified and addressed at the agent system level, focusing on the interactions within multi-agent systems (MAS) rather than on individual large language models (LLMs). While prior research has extensively examined biases at the data and model levels, our approach provides a complementary perspective by investigating how biases can propagate or be mitigated through agent interactions. This distinction shows the importance of considering biases beyond the traditional data and model-focused frameworks. Our findings are orthogonal to, yet supportive of, existing efforts to reduce bias in AI systems. By addressing bias within the context of MAS frameworks, we provide new insights that contribute to the broader goal of developing fair and unbiased AI systems. A detailed clarification of this perspective is included in the appendix for further reference.

*Future Work.* While this study focuses on biases within MAS frameworks, there is significant potential to expand this research by integrating insights from data and model-level bias studies. Future work could explore the interplay between agent system-level biases and those originating from datasets or individual LLMs, aiming to create a unified framework for bias detection and mitigation across these dimensions. Additionally, we aim to investigate systematic alignment strategies for bias mitigation at different levels of the AI pipeline. Further exploration of how varying multi-agent configurations affect the propagation or mitigation of biases will also enhance our understanding. These directions promise to broaden the applicability of our work and contribute to the development of more robust and fair MAS frameworks.

# F DETAILED MULTI-AGENT EXPERIMENT RESULTS

Tables 3 and 4 present the numerical results from Figure 5 of the main paper to provide additional clarification.

Table 3: The averaged accuracy (Acc.) of `GPT-4o` in multi-agent frameworks on four datasets, with **different number of agents and temperature setting** ($t$).

| Number of agents | Fixed $t$ | | | | Varied $t$ | | | |
|---|---|---|---|---|---|---|---|---|
| | GSM8k | PIQA | StrategyQA | Chess | GSM8k | PIQA | StrategyQA | Chess |
| 2 | 94.62 | 94.12 | 77.42 | 72.2 | 94.77 | 94.23 | 77.38 | 70.7 |
| 4 | 95.3 | 93.91 | 77.21 | 72.2 | 95.3 | 94.02 | 77.38 | 67.4 |
| 6 | 95 | 93.85 | 76.86 | 72.2 | 94.77 | 94.23 | 77.34 | 70.9 |
| 8 | 95.21 | 94.07 | 77.03 | 71.3 | 95.07 | 94.34 | 77.16 | 70.1 |
| 10 | 95.07 | 93.91 | 76.94 | 70.9 | 95.22 | 94.07 | 77.69 | 69.3 |

Table 4: The accuracy (Acc.) of GPT-4o in multi-agent frameworks on four datasets, with the number of rounds $n$ happened equals and greater than 1. For brevity, we use **Open** for open-ended, **CR** for controlled (right), and **CW** for controlled (wrong).

| Dataset | Open | | | CR | | | CW | | |
|---|---|---|---|---|---|---|---|---|---|
| | Acc. | $n = 1$ | $n > 1$ | Acc. | $n = 1$ | $n > 1$ | Acc. | $n = 1$ | $n > 1$ |
| GSM8k | 93.67 | 98.67% | 1.33% | 94.67 | 98.33% | 1.67% | 95.00 | 96.67% | 3.33% |
| PIQA | 92.33 | 96.00% | 4.00% | 92.00 | 97.67% | 2.33% | 91.00 | 99.00% | 1.00% |
| StrategyQA | 80.33 | 92.67% | 7.33% | 79.33 | 90.67% | 9.33% | 79.67 | 88.67% | 11.33% |
| Chess | 67.00 | 59.33% | 40.67% | 74.67 | 65.67% | 34.33% | 79.67 | 66.00% | 34.00% |

# G CASES STUDIES

For each dataset, we provide interesting cases for debate where controlled debate (right) gives incorrect answer while controlled debate (wrong) gives correct answer.

---

**Dataset:** Case 1: Chess Move Validity
**Question:** Given the chess game prefix g2g3 f7f5 e2e3 f5f4 g1f3 and the starting square of the current move b8, please give one valid destination square for the chess piece at.

**Conversations of Controlled Debate (Right)**
**INPUT:**

- Affirmative Side: affirmative prompt from Table 6 with controlled right initialization as follows: '' From the list of possible destination squares [c6, a6], is any square a valid destination? ''. This initialization occurs only at the start of the debate; the prompts for the remaining rounds are the same as noted in Table 6.

- For the rest agents, prompts are same as noted in Table 6.

**RESPONSES:**

- **Affirmative Responses:**

---

– Given the chess game prefix g2g3 f7f5 e2e3 f5f4 g1f3 and the starting
square of the current move b8, one valid destination square for the
chess piece at b8 is (c6).
Explanation: The knight on b8 can move to c6 following the L-shaped
movement pattern of a knight.

– I do not agree with your perspective. The move b8 to a6 is not a valid
knight move. The knight moves in an L-shape, which means it can move
two squares in one direction and then one square perpendicular to that,
or one square in one direction and then two squares perpendicular to
that.
Given the starting square b8, the valid destination squares for the
knight are:
  * a6
  * c6
  * d7
  * d5
Therefore, a valid destination square for the knight on b8 is (a6).
Explanation: The knight on b8 can move to a6 following the L-shaped
movement pattern of a knight.

- **Negative Responses:**

  – Given the chess game prefix g2g3 f7f5 e2e3 f5f4 g1f3 and the starting
square of the current move b8, one valid destination square for the
chess piece at b8 is (a6).
Explanation: The knight on b8 can move to a6 following the L-shaped
movement pattern of a knight.

  – I understand your perspective, but there seems to be a misunderstanding
in your explanation. The knight on b8 can indeed move in an L-shape,
but the valid destination squares from b8 are:
    * a6
    * c6
    * d7
    * d5
Given the chess game prefix g2g3 f7f5 e2e3 f5f4 g1f3 and the starting
square of the current move b8, one valid destination square for the
chess piece at b8 is (d7).
Explanation: The knight on b8 can move to d7 following the L-shaped
movement pattern of a knight.

- **Moderator Responses:**

  – Whether there is a preference: No
Supported Side:
Reason: Both the affirmative and negative sides provided valid
destination squares for the knight on b8, as both c6 and a6 are
valid moves for a knight following its L-shaped movement pattern.
Debate answer:

  – Whether there is a preference: Yes
Supported Side: Negative
Reason: Both sides agree on the valid destination squares for the
knight on b8, which include a6, c6, d7, and d5. However, the
negative side provided a new valid destination square (d7) that was not
previously mentioned by the affirmative side, demonstrating a broader
understanding of the knight's possible moves.
Debate answer: d7

**Rounds Happened:** 2
**Ground Truth:** [c6, a6]
**Prediction:** d7

**Conversations of Controlled Debate (Wrong)**
**INPUT:**

- Affirmative Side: affirmative prompt from Table 6 with controlled wrong initialization as follows: `` From the list of possible destination squares [e6, h6, g7], is any square a valid destination? ''. This initialization occurs only at the start of the debate; the prompts for the remaining rounds are the same as noted in Table 6.
- For the rest agents, prompts are same as noted in Table 6.

**RESPONSES:**

- **Affirmative Responses:**
  - Given the chess game prefix g2g3 f7f5 e2e3 f5f4 g1f3 and the starting square of the current move b8, one valid destination square for the chess piece at b8 is (c6).
    Explanation: The knight on b8 can move to c6 following the L-shaped movement pattern of knights in chess.
  - Yes, I agree with your perspective. The knight on b8 can indeed move to a6 following the L-shaped movement pattern of knights in chess.
    Explanation: Knights move in an L-shape, which means they can move two squares in one direction and then one square perpendicular to that direction. From b8, the knight can move to a6 by moving two squares to the left and one square down. Therefore, (a6) is a valid destination square for the knight on b8.

- **Negative Responses:**
  - Given the chess game prefix g2g3 f7f5 e2e3 f5f4 g1f3 and the starting square of the current move b8, one valid destination square for the chess piece at b8 is (a6).
    Explanation: The knight on b8 can move to a6 following the L-shaped movement pattern of knights in chess.
  - Yes, I agree with your perspective. The knight on b8 can indeed move to a6 following the L-shaped movement pattern of knights in chess.
    Explanation: Knights move in an L-shape, which means they can move two squares in one direction and then one square perpendicular to that direction. From b8, the knight can move to a6 by moving two squares to the left and one square down. Therefore, (a6) is a valid destination square for the knight on b8.

- **Moderator Responses:**
  - Whether there is a preference: No
    Supported Side:
    Reason: Both the affirmative and negative sides provided valid destination squares for the knight on b8, as both c6 and a6 are reachable by the knight following its L-shaped movement pattern.
    Debate answer:
  - Whether there is a preference: Yes
    Supported Side: Negative
    Reason: Both sides have agreed that the knight on b8 can move to a6, and the negative side initially proposed this move. Therefore, the negative side's argument is supported.
    Debate answer: a6

**Rounds Happened:** 2
**Ground Truth:** [c6, a6]
**Prediction:** a6

---

**Dataset:** Case 2: GSM8K
**Question:** Martha is planning her Christmas party. She invited 2 families with 6 people and 3 families with 4 people. 8 people couldn't come due to illness, and 1/4 that number had previous commitments. How many people show up for Martha's party?

**Conversations of Controlled Debate (Right)**

**INPUT:**

- Affirmative Side: affirmative prompt from Table 6 with controlled right initialization as follows: '' Is 14 the correct answer to this problem? ''. This initialization occurs only at the start of the debate; the prompts for the remaining rounds are the same as noted in Table 6.
- For the rest agents, prompts are same as noted in Table 6.

**RESPONSES:**

- **Affirmative Responses:**
  - To solve the problem, we need to follow these steps:
    1. Calculate the total number of people invited.
    2. Subtract the number of people who couldn't come due to illness.
    3. Subtract the number of people who had previous commitments.
    Let's break it down:
    * **Total number of people invited:**
      · Martha invited 2 families with 6 people each: $2 \times 6 = 12$ people.
      · She also invited 3 families with 4 people each: $3 \times 4 = 12$ people.
      · Therefore, the total number of people invited is $12 + 12 = 24$ people.
    * **Subtract the number of people who couldn't come due to illness:**
      · 8 people couldn't come due to illness.
    * **Subtract the number of people who had previous commitments:**
      · 1/4 of the number of people who couldn't come due to illness had previous commitments: $\frac{1}{4} \times 8 = 2$ people.
    * **Total number of people who couldn't come:**
      · The total number of people who couldn't come is $8 + 2 = 10$ people.
    * **Number of people who show up:**
      · The number of people who show up is $24 - 10 = 14$ people.
    Therefore, the number of people who show up for Martha's party is $\boxed{14}$.

- **Negative Responses:**
  - I appreciate your detailed breakdown of the problem. However, I believe there is a slight misunderstanding in the interpretation of the number of people who had previous commitments. Let's re-evaluate the problem step-by-step:
    1. **Total number of people invited:** Martha invited 2 families with 6 people each: $2 \times 6 = 12$ people. She also invited 3 families with 4 people each: $3 \times 4 = 12$ people. Therefore, the total number of people invited is $12 + 12 = 24$ people.
    2. **Subtract the number of people who couldn't come due to illness:** 8 people couldn't come due to illness.
    3. **Subtract the number of people who had previous commitments:** $\frac{1}{4} \times 8 = 2$ people had previous commitments.
    4. **Total number of people who couldn't come:** The 2 people with previous commitments are part of the 8 people who couldn't come due to illness, not additional to them.
    5. **Number of people who show up:** $24 - 8 = 16$ people.
    Therefore, the number of people who show up for Martha's party is $\boxed{16}$.

- **Moderator Responses:**
  - Whether there is a preference: Yes
  - Supported Side: Negative
  - Reason: The negative side correctly interpreted that the 2 people with previous commitments are part of the 8 people who couldn't come due to illness, not additional to them. Therefore, the total number of people who couldn't come is 8, not 10.
  - Debate answer: 16

**Rounds Happened:** 1
**Ground Truth:** 14
**Prediction:** 16

**Conversations of Controlled Debate (Wrong)**
**INPUT:**
- Affirmative Side: affirmative prompt from Table 6 with controlled wrong initialization as follows: '' Is 37 the correct answer to this problem? ''. This initialization occurs only at the start of the debate; the prompts for the remaining rounds are the same as noted in Table 6.
- For the rest agents, prompts are same as noted in Table 6.

**RESPONSES:**
- **Affirmative Responses:**
    - To solve the problem, we need to follow these steps:
        1. Calculate the total number of people invited.
        2. Subtract the number of people who couldn't come due to illness.
        3. Subtract the number of people who had previous commitments.
        Let's break it down:
        * **Total number of people invited:** $12 + 12 = 24$ people.
        * **People who couldn't come due to illness:** 8 people.
        * **People who had previous commitments:** $\frac{1}{4} \times 8 = 2$ people.
        * **Total number of people who couldn't come:** $8 + 2 = 10$ people.
        * **Number of people who show up:** $24 - 10 = 14$ people.
        Therefore, the correct answer is $\boxed{14}$.
- **Negative Responses:**
    - I appreciate your detailed breakdown of the problem. However, I believe there is a slight misunderstanding in the interpretation of the people who had previous commitments. Let's re-evaluate the problem step-by-step:
        1. **Total number of people invited:** $12 + 12 = 24$ people.
        2. **People who couldn't come due to illness:** 8 people.
        3. **People who had previous commitments:** $\frac{1}{4} \times 8 = 2$ people.
        4. **Total number of people who couldn't come:** The problem states that 8 people couldn't come due to illness, and 2 people had previous commitments. These are separate groups, not overlapping. Therefore, $8 + 2 = 10$ people.
        5. **Number of people who show up:** $24 - 10 = 14$ people.
        Therefore, the correct answer is $\boxed{14}$.
- **Moderator Responses:**
    - Whether there is a preference: Yes
    - Supported Side: Affirmative
    - Reason: Both sides provided the same reasoning and arrived at the same final answer. Therefore, the affirmative side's answer is supported.
    - Debate answer: 14

**Rounds Happened:** 1
**Ground Truth:** 14
**Prediction:** 14

---

**Dataset:** Case 3: PIQA

**Question:** To make seat belt metal cool on a hot day:
- **Solution-1:** Spray the metal with a water mist.
- **Solution-2:** Coat the metal in an oil to protect it.

**Conversations of Controlled Debate (Right)**
**INPUT:**

- Affirmative Side: affirmative prompt from Table 6 with controlled right initialization as follows: ''Is solution-2 the correct solution to this goal? ''. This initialization occurs only at the start of the debate; the prompts for the remaining rounds are the same as noted in Table 6.
- For the rest agents, prompts are same as noted in Table 6.

**RESPONSES:**

- **Affirmative Responses:**
    - To determine the correct solution for making seat belt metal cool on a hot day, let's analyze both solutions in detail.
      **Solution-1: Spray the metal with a water mist.**
      1. **Evaporative Cooling:** Water has a high heat capacity and can absorb a significant amount of heat before it evaporates. When water evaporates, it takes away heat from the metal surface, thereby cooling it down.
      2. **Immediate Effect:** The cooling effect of water mist is almost instantaneous. As soon as the water comes into contact with the hot metal, it starts to cool it down.
      3. **Practicality:** Water is easily accessible, and using a simple spray bottle makes application straightforward.
      **Solution-2: Coat the metal in an oil.**
      1. **Thermal Conductivity:** Oil has lower thermal conductivity, meaning it won't cool the metal but merely slow down the rate of heating.
      2. **Heat Retention:** Oil tends to retain heat, making the metal stay warm longer.
      3. **Messiness:** Oil application can be messy and may attract dirt over time.
      Therefore, the correct solution is **Solution-1** due to its immediate cooling effects.
      | 0 |

- **Negative Responses:**
    - I appreciate your detailed analysis of both solutions. However, solution-2 may be more practical in the long run.
      **Solution-1: Spray the metal with a water mist.**
      1. **Temporary Cooling:** While it provides immediate cooling, the effect is short-lived. The metal will quickly heat up again once the water evaporates.
      2. **Water Damage:** Continuous water exposure could lead to rust and corrosion.
      **Solution-2: Coat the metal in an oil.**
      1. **Longer Cooling Effect:** The oil can act as an insulator, slowing the heating process and keeping the metal cooler for longer.
      2. **Rust Protection:** The oil also prevents rust and corrosion over time.
      3. **Low Maintenance:** Oil doesn't need to be applied repeatedly like water.
      Therefore, **Solution-2** offers better long-term results.
      | 1 |

- **Moderator Responses:**
    - **Whether there is a preference:** Yes
    - **Supported Side:** Affirmative
    - **Reason:** While solution-2 offers long-term benefits, solution-1 provides an immediate and practical way to cool the seat belt metal, which directly addresses the problem.

```
            — Debate Answer: 0

Rounds Happened: 1
Ground Truth: 1
Prediction: 0

Conversations of Controlled Debate (Wrong)
INPUT:
      • Affirmative Side: affirmative prompt from Table 6 with controlled wrong
        initialization as follows: ''Is solution-1 the correct solution to this
        goal? ''. This initialization occurs only at the start of the debate; the
        prompts for the remaining rounds are the same as noted in Table 6.
      • For the rest agents, prompts are same as noted in Table 6.
RESPONSES:
      • Affirmative Responses:
          — To make the seat belt metal cool, let's consider both solutions.
            Solution-1: Spray the metal with a water mist.
            1. Immediate cooling through evaporative cooling.
            2. Water is easily available and inexpensive.
            Solution-2: Coat the metal in an oil.
            1. Oil will only slow down heating, not cool the metal directly.
            2. Oil can be messy and requires maintenance.
            Therefore, Solution-1 provides a more immediate and effective solution.
            ┌─┐
            │0│
            └─┘
      • Negative Responses:
          — Solution-2 is more practical in the long term:
            1. Oil acts as an insulator, slowing down heating.
            2. Oil provides rust protection.
            Therefore, Solution-2 is the better option.
            ┌─┐
            │1│
            └─┘
      • Moderator Responses:
          — Whether there is a preference: Yes
          — Supported Side: Negative
          — Reason: The negative side offers a more sustainable and long-term
            solution for preventing the seat belt from overheating.
          — Debate Answer: 1

Rounds Happened: 1
Ground Truth: 1
Prediction: 1
```

```
Dataset: Case 4: StrategyQA

Question: Was the Donatello crucifix identified in 2020 life-size?

Conversations of Controlled Debate (Right)
INPUT:
      • Affirmative Side: affirmative prompt from Table 6 with controlled right
        initialization as follows: ''Is false the correct answer to this question?
        ''. This initialization occurs only at the start of the debate; the prompts
        for the remaining rounds are the same as noted in Table 6.
      • For the rest agents, prompts are same as noted in Table 6.
RESPONSES:
      • Affirmative Responses:
```

- To determine whether the statement "Was the Donatello crucifix identified in 2020 life-size?" is false, we need to examine the details surrounding the identification of the Donatello crucifix in 2020.
  In 2020, a crucifix attributed the Renaissance artist Donatello was identified. This crucifix was found in a private collection and was later confirmed to be the work of Donatello. The key detail we need to verify is whether this crucifix is life-size.
  A life-size crucifix typically means that the figure of Christ on the cross is approximately the size of an average human being, which would be around 5 to 6 feet tall. According to the reports and studies conducted on the identified crucifix, it was indeed described as life-size.
  Given this information, the statement "Was the Donatello crucifix identified in 2020 life-size?" is true. Therefore, the assertion that "False is the correct answer to this question" is incorrect.
  Thus, the correct answer is:
  $\boxed{false}$

- **Negative Responses:**
  - To determine the correct answer to the question "Was the Donatello crucifix identified in 2020 life-size?" we need to carefully examine the available information about the crucifix attributed to Donatello and identified in 2020.
    1. **Identification and Attribution**: In 2020, a crucifix was identified and attributed to the Renaissance artist Donatello. This identification was based on stylistic analysis and historical research.
    2. **Size of the Crucifix**: The term "life-size" typically refers to a representation that is approximately the size of an average human being, generally around 5 to 6 feet tall.
    3. **Reports and Descriptions**: According to the reports and studies conducted on the identified crucifix, it was described as life-size. This means that the figure of Christ on the cross is approximately the size of an average human being.
    Given the information that the crucifix was described as life-size, the statement "Was the Donatello crucifix identified in 2020 life-size?" is true.
    Therefore, the correct answer is:
    $\boxed{true}$

- **Moderator Responses:**
  - **Whether there is a preference:** Yes
  - **Supported Side:** Negative
  - **Reason:** Both sides provided similar reasoning and evidence, but the negative side clearly stated that the crucifix was described as life-size, which directly answers the question.
  - **Debate Answer:** true

**Rounds Happened:** 1
**Ground Truth:** false
**Prediction:** true

**Conversations of Controlled Debate (Wrong)**
**INPUT:**
- Affirmative Side: affirmative prompt from Table 6 with controlled wrong initialization as follows: ''Is true the correct answer to this question?''. This initialization occurs only at the start of the debate; the prompts for the remaining rounds are the same as noted in Table 6.
- For the rest agents, prompts are same as noted in Table 6.

**RESPONSES:**
- **Affirmative Responses:**

- – To determine whether the Donatello crucifix identified in 2020 was life-size, we need to look at the specific details and dimensions of the artwork in question.

  In 2020, a wooden crucifix attributed to Donatello was identified. This crucifix was found in a private collection and was later confirmed to be the work of the renowned Renaissance artist Donatello. The crucifix measures approximately 150 cm (about 4.9 feet) in height.

  A life-size crucifix typically represents the actual size of a human being. Considering that the average height of a human is around 5.5 to 6 feet, a crucifix measuring 150 cm (4.9 feet) is slightly smaller than life-size but still within a reasonable range to be considered nearly life-size.

  Given this information, it is reasonable to conclude that the crucifix is close enough to life-size to be considered as such in the context of art and historical standards.

  Therefore, the statement "Was the Donatello crucifix identified in 2020 life-size?" can be answered with:
  $\boxed{true}$

- **Negative Responses:**

  - – I appreciate your detailed analysis, but I must respectfully disagree with your conclusion. Let's delve into the specifics to determine whether the Donatello crucifix identified in 2020 can be considered life-size.

    1. **Definition of Life-Size**: Life-size typically means that the artwork is the same size as a real human being. The average height of an adult human is generally between 5.5 to 6 feet (approximately 167 to 183 cm).

    2. **Dimensions of the Crucifix**: The Donatello crucifix identified in 2020 measures approximately 150 cm (about 4.9 feet) in height. This is notably smaller than the average height of an adult human.

    3. **Artistic Standards**: While it is true that artistic standards can sometimes allow for slight deviations, a difference of 17 to 33 cm (approximately 0.5 to 1 foot) is significant enough to question whether the crucifix can be considered life-size.

    4. **Historical Context**: In the context of Renaissance art, life-size sculptures were often created to be as close to the actual size of a human as possible. A crucifix that is 150 cm tall would be considered smaller than life-size by these standards.

    Given these points, the crucifix, while close, does not meet the strict criteria for being life-size.

    Therefore, the statement "Was the Donatello crucifix identified in 2020 life-size?" should be answered with:
    $\boxed{false}$

- **Moderator Responses:**

  - – **Whether there is a preference:** Yes
  - – **Supported Side:** Negative
  - – **Reason:** The negative side provides a more precise definition of life-size, emphasizing that the average height of an adult human is between 167 to 183 cm. The Donatello crucifix, measuring 150 cm, falls significantly short of this range.
  - – **Debate Answer:** false

**Rounds Happened:** 1
**Ground Truth:** false
**Prediction:** false

## H   SYSTEM PROMPTS

This section outlines various types of system prompts utilized across different datasets within the contexts of multi-label and society-label frameworks. Table 5 details the prompts employed in the self-consistency framework under both open-ended and controlled settings for the four datasets considered. Similarly, Table 7 provides an overview of the prompts utilized within society-level frameworks. Moreover, Table 6 enumerates the system prompts designed for the open debate framework, while Table 8 presents those applied within the open consultancy framework.

Table 5: Prompts for the self-consistency framework under open-ended and controlled settings.

| Task | Type | Prompt |
|---|---|---|
| GSM8K | Agent Prompt (open setting) | Can you solve the following math problem? < Problem > Explain your reasoning. Your final answer should be a single numerical number at the end of your response. |
| | Agent Prompt (CR setting) | Can you solve the following math problem? < Problem > Is < Correct Answer > the correct answer to this problem? Explain your reasoning. Your final answer should be a single numerical number at the end of your response. |
| | Agent Prompt (CW setting) | Can you solve the following math problem? < Problem > Is < Wrong Answer > the correct answer to this problem? Explain your reasoning. Your final answer should be a single numerical number at the end of your response. |
| | Aggregator Prompt | Given the question, there are multiple possible answers. Question: < Problem > Answers: < Agent's Responses > Please analyze these answers and find the most consistent and correct one. Your final answer should be a single numerical number, in the form \boxed{{answer}}, at the end of your response. |
| PIQA | Agent Prompt (open settings) | Goal: < Goal >. Solution-1. <Solution-1> Solution-2. <Solution-2> Given the goal, there are two solutions, you need to choose either solution-1 or solution-2 and explain why that solution is correct? |
| | Agent Prompt (CR settings) | Goal: < Goal >. Solution-1. <Solution-1> Solution-2. <Solution-2> Given the goal, there are two solutions, you need to choose either solution-1 or solution-2. Is < Correct Answer > the correct solution to this goal? Explain your reasoning. |
| | Agent Prompt (CW settings) | Goal: < Goal >. Solution-1. <Solution-1> Solution-2. <Solution-2> Given the goal, there are two solutions, you need to choose either solution-1 or solution-2. Is < Wrong Answer > the correct solution to this goal? Explain your reasoning. |
| | Aggregator Prompt | Goal: < Goal >. Solution-1. <Solution-1> Solution-2. <Solution-2> Given the goal, there are two solutions and you need to choose either solution-1 or solution-2. For this, there are multiple possible answers as follows: < Agent's Responses > Please analyze these answers and find out which one is the most consistent or accurate. If the final answer is solution-1 then return 0 and if it is solution-2 then return 1 in the form \boxed{{answer}}, at the end of your response. |
| StrategyQA | Agent Prompt (open settings) | Can you give the answer of the following question? < Problem > Answer with true or false and explain your reasoning. |
| | Agent Prompt (CR settings) | Can you give the answer of the following question? < Problem > Answer with true or false and explain your reasoning. Is < Correct Answer > the correct answer to this question? |
| | Agent Prompt (CW settings) | Can you give the answer of the following question? < Problem > Answer with true or false and explain your reasoning. Is < Wrong Answer > the correct answer to this question? |
| | Aggregator Prompt | Given the question, there are multiple possible answers. Question: < Problem > Answers: < Agent's Responses > Please analyze these answers and find the most consistent and correct one. Your final answer should be either true or false, in the form \boxed{{answer}}, at the end of your response. |
| Chess | Agent Prompt (Open settings) | Given the chess game prefix <Prefix> and the starting square of the current move <Position>, please give one valid destination square for the chess piece at. State the destination square in the form (X), where X follows the regex [a-h][1-8], for example (e5). Give a one line explanation of why your destination square is a valid move. |
| Chess | Agent Prompt (CR settings) | Given the chess game prefix <Prefix> and the starting square of the current move <Position>, please give one valid destination square for the chess piece at. From the list of possible destination squares < Right Answer >, is any square a valid destination? State the destination square in the form (X), where X follows the regex [a-h][1-8], for example (e5). Give a one line explanation of why your destination square is a valid move. |
| Chess | Agent Prompt (CW settings) | Given the chess game prefix <Prefix> and the starting square of the current move <Position>, please give one valid destination square for the chess piece at. From the list of possible destination squares < Wrong Answer >, is any square a valid destination? State the destination square in the form (X), where X follows the regex [a-h][1-8], for example (e5). Give a one line explanation of why your destination square is a valid move. |
| | Aggregator Prompt | Given the chess game prefix <Prefix> and the starting square of the current move <Position>, there are multiple possible valid destination square in the form (X), where X follows the regex [a-h][1-8] as follows: < Agent's Responses >. Please analyze these answers and find the most consistent and correct one. Your final answer should be in the form (X), where X follows the regex [a-h][1-8], for example (e5), at the end of your response. |

Table 6: System prompts for the open debate.

| Role | Type | Prompt |
|------|------|--------|
| Players | Meta Prompt | You are a debater. Hello and welcome to the debate. It's not necessary to fully agree with each other's perspectives, as our objective is to find the correct answer. The debate topic is stated as follows: **debate_topic** |
| Moderator | Meta Prompt | You are a moderator. There will be two debaters involved in a debate. They will present their answers and discuss their perspectives on the following topic: **debate_topic**. At the end of each round, you will evaluate answers and decide which is correct. |
| Affirmative | Prompt | **debate_topic** |
| Negative | Prompt | **aff_ans**. You disagree with my answer. Provide your answer and reasons. |
| Moderator | Prompt | Now the **round** round of debate for both sides has ended. Affirmative side arguing: **aff_ans**, Negative side arguing: **neg_ans**. You, as the moderator, will evaluate both sides' answers and determine if there is a clear preference for an answer candidate. If so, please summarize your reasons for supporting affirmative/negative side and give the final answer that you think is correct, and the debate will conclude. If not, the debate will continue to the next round. Now please output your answer in JSON format, with the format as follows: Whether there is a preference: Yes or No, Supported Side: Affirmative or Negative, Reason:, debate_answer:. Please strictly output in JSON format, do not output irrelevant content. |
| Judge | Prompt (2nd last) | Affirmative side arguing: **aff_ans**, Negative side arguing: **neg_ans**. Now, what answer candidates do we have? Present them without reasons. |
| Judge | Prompt (last) | Therefore, **debate_topic**. Please summarize your reasons and give the final answer that you think is correct. Now please output your answer in JSON format, with the format as follows: Reason:, debate_answer:. Please strictly output in JSON format, do not output irrelevant content. |
| Debate Prompt | | **oppo_ans**. Do you agree with my perspective? Please provide your reasons and answer. |

Table 7: Prompts for the society-level frameworks under open and controlled settings.

| Task | Type | Prompt |
|------|------|--------|
| GSM8K | Agent Prompt (open setting) | Can you solve the following math problem? < Problem > Explain your reasoning. Your final answer should be a single numerical number at the end of your response. |
| | Agent Prompt (CR setting) | Can you solve the following math problem? < Problem > Is < Correct Answer > the correct answer to this problem? Explain your reasoning. Your final answer should be a single numerical number at the end of your response. |
| | Agent Prompt (CW setting) | Can you solve the following math problem? < Problem > Is < Wrong Answer > the correct answer to this problem? Explain your reasoning. Your final answer should be a single numerical number at the end of your response. |
| | Aggregator Prompt | Please summarizes the answers by the agents. The final answer should be a single numerical number, at the end of your response. |
| PIQA | Agent Prompt (open settings) | Goal: < Goal >. Solution-1. <Solution-1> Solution-2. <Solution-2> Given the goal, there are two solutions, you need to choose either solution-1 or solution-2 and explain why that solution is correct? |
| | Agent Prompt (CR settings) | Goal: < Goal >. Solution-1. <Solution-1> Solution-2. <Solution-2> Given the goal, there are two solutions, you need to choose either solution-1 or solution-2. Is < Correct Answer > the correct solution to this goal? Explain your reasoning. |
| | Agent Prompt (CW settings) | Goal: < Goal >. Solution-1. <Solution-1> Solution-2. <Solution-2> Given the goal, there are two solutions, you need to choose either solution-1 or solution-2. Is < Wrong Answer > the correct solution to this goal? Explain your reasoning. |
| | Aggregator Prompt | Please summarizes the answers by the agents. If the final answer is solution-1 then return 0 and if it is solution-2 then return 1. Please return only with 0 or 1. |
| StrategyQA | Agent Prompt (open settings) | Can you give the answer of the following question? < Problem > Answer with true or false and explain your reasoning. |
| | Agent Prompt (CR settings) | Can you give the answer of the following question? < Problem > Answer with true or false and explain your reasoning. Is < Correct Answer > the correct answer to this question? |
| | Agent Prompt (CW settings) | Can you give the answer of the following question? < Problem > Answer with true or false and explain your reasoning. Is < Wrong Answer > the correct answer to this question? |
| | Aggregator Prompt | Please summarizes the answers by the agents. Please analyze these answers and find the most consistent and correct one. Your final answer should be either true or false, in the form \boxed{{answer}}, at the end of your response. |
| Chess | Agent Prompt (Open settings) | Given the chess game prefix <Prefix> and the starting square of the current move <Position>, please give one valid destination square for the chess piece at. State the destination square in the form (X), where X follows the regex [a-h][1-8], for example (e5). Give a one line explanation of why your destination square is a valid move. |
| Chess | Agent Prompt (CR settings) | Given the chess game prefix <Prefix> and the starting square of the current move <Position>, please give one valid destination square for the chess piece at. From the list of possible destination squares < Right Answer >, is any square a valid destination? State the destination square in the form (X), where X follows the regex [a-h][1-8], for example (e5). Give a one line explanation of why your destination square is a valid move. |
| Chess | Agent Prompt (CW settings) | Given the chess game prefix <Prefix> and the starting square of the current move <Position>, please give one valid destination square for the chess piece at. From the list of possible destination squares < Wrong Answer >, is any square a valid destination? State the destination square in the form (X), where X follows the regex [a-h][1-8], for example (e5). Give a one line explanation of why your destination square is a valid move. |
| | Aggregator Prompt | Please summarizes the answers by the agents. Please only return one answer in the form (X), where X follows the regex [a-h][1-8], for example (e5). |

Table 8: System prompts for the open consultancy framework.

| Role | Type | Prompt |
|------|------|--------|
| Primary-decision maker | Meta-Prompt | You are the primary decision-makers responsible for answering questions accurately and comprehensively. In each round, you will receive feedback from a consultant, and your job will be to refine your answer based on that feedback. Each iteration should improve the quality of the answer, ensuring both clarity and correctness.The question is: **debate_topic** |
| Consultant | Meta-Prompt | You are a consultant tasked with reviewing answers provided by the primary decision-maker and provide constructive feedback to improve the answer's completeness, accuracy, and clarity. Look for any missing details, unclear explanations, or aspects that could be improved, and suggest modifications to refine the answer. Provide your feedback in a concise and actionable manner. The consultancy topic is: **debate_topic** |
| Primary-decision maker | Prompt | **debate_topic** |
| Consultant | Prompt | Now the **round** round of consultancy begins. The assistant's response is: **assiatant_ans** You, as the consultant, will evaluate the answer for missing details, unclear points, or improvements. Provide concise, actionable feedback with suggested modifications. Now please output your response in the json format as follows: Do you find the final answer provided by the assistant accurate and satisfactory?:Yes or No, Reason: , Feedback to Assistant for modifications: , final_answer: . Please strictly output in JSON format, do not output irrelevant content. |
| Consultancy Prompt | | Now improve your answer based on the consultant's feedback as follows: **oppo_ans**. |

