# OpenReview forum: "Understanding Prejudice and Fidelity of Diverge-to-Converge Multi-Agent Systems"
_ICLR.cc/2025/Conference — Submitted to ICLR 2025_

### Official Review · Reviewer_xVhp · 2024-10-27

**Soundness:** 3
**Presentation:** 3
**Contribution:** 2
**Rating:** 6
**Confidence:** 4

**Summary:**

This paper focuses on the Diverge-to-Converge (D2C) frameworks and highlights the challenges of prejudice and fidelity in D2C frameworks. The authors define prejudice and fidelity as the performance variation under changed conditions and scaling laws, respectively. To evaluate prejudice and fidelity, this paper introduces APF-Bench using the proposed Dataset Refinement. The results confirm the findings.

**Strengths:**

1. The paper is well-structured and easy to follow. The inclusion of informative figures and tables enhances clarity.
2. This paper reveals two key challenges: the impact of initial conditions and the number of agents on the final performance.
3. The experiments span many task-domains and multiple models.

**Weaknesses:**

1. This paper mainly proposes a benchmark to test and validate these challenges rather than further addressing them.
2. In more complex scenarios, problem reframing is difficult.

**Questions:**

Minor comments:
1. In Figure 1, should the question be "A ship travels 80 miles east/west and 150 miles north. How far is the ship from its starting point?".
2. D2C instead of C2D, e.g. Section 5 Debatepedia, Dataset Problem Reframing, etc.
3. Inconsistent symbol representation. In Section 3, C stands for the total number of calls, whereas in Section 4, C stands for Agent Count.

---

> ### Author Response · Authors · 2024-11-23
>
> > W1: This paper mainly proposes a benchmark to test and validate these challenges rather than further addressing them.
>
> **Comment**:
> While the APF benchmark is a significant contribution of this paper, we would like to emphasize the actionable strategies proposed and linked with experiments to address prejudice and fidelity challenges. For example:
>
> - **Mitigating Confirmation Bias**:
>   We introduce problem reframing with controlled initialization (right or wrong solutions) for both model-label and society-label frameworks `[Lines: 419–442]`. `Table 1` showcases three settings: open (vanilla D2C framework), controlled (right), and controlled (wrong) initialization. Notably, for a complex task like Chess Move Validity on GPT-4o, we achieve a 9.7% performance improvement over the open-ended framework `[Table 1]`. Similarly, `Table 2 (Appendix)` provides performance data across four models and three settings, reinforcing the superiority of the problem reframing strategy.
>
> - **Judge Bias Ratio**:
>   Our analysis explores how initial agent roles affect "judge bias ratio" in frameworks like Debate. By quantifying bias ratios `[Lines: 444–457, Figure 3]`, we show that controlled debate settings significantly decrease affirmative bias influence, particularly for PIQA and StrategyQA, while increasing negative bias. This shift highlights the impact of control mechanisms during debates on judgment formation `[Lines: 444–457]`. In the revised version, we also propose a potential solution to address social biases in D2C frameworks.
>
> - **Exploring Fidelity**:
>   We analyzed scaling behavior in agent interactions and resource usage `[Section 6.2]`. Key findings include:
>   1. **Improved Scaling**: Complex tasks like Chess Move Validity benefit significantly from scaling resources. `[Figure 5(a), Lines: 496–510]`
>   2. **Saturation Effects**: Simpler tasks like PIQA show performance plateaus after four agents, indicating diminishing returns. `[Lines: 504–510]`
>   3. **Trade-offs in Agent Interactions**: Excessive scaling of interaction rounds leads to degraded performance due to coordination complexity, especially in society-level frameworks. `[Lines: 510–517, Figure 6]`
>
> These insights and solutions enhance fairness and robustness in multi-agent interactions, offering practical guidance for real-world applications and valuable tools for refining such systems beyond benchmarking.
>
> ---
>
> > W2: In more complex scenarios, problem reframing is difficult.
>
> **Comment**:  Replied in the general response.
>
> ---
>
>
> > Q1: In Figure 1, should the question be "A ship travels 80 miles east/west and 150 miles north. How far is the ship from its starting point?"
>
> **Comment**:
> Thank you for bringing this to our attention. The question has been updated in the revised version.
>
> ---
>
> > Q2: D2C instead of C2D, e.g., Section 5 Debatepedia, Dataset Problem Reframing, etc.
>
> **Comment**:
> Thank you for pointing this out. We have corrected the references to "D2C" in the revised version. Please refer to `lines 287 and 295`.
>
> ---
>
> > Q3: Inconsistent symbol representation. In Section 3, C stands for the total number of calls, whereas in Section 4, C stands for Agent Count.
>
> **Comment**:
> Thank you for highlighting this inconsistency. The symbols have been unified in the revised version.

---

> > ### Comment · Reviewer_xVhp · 2024-11-26
> >
> > Thank you for clarifying the contribution and I have raised my score.

---

> ### Author Response · Authors · 2024-11-27
> **Thank You to Reviewer xVhp**
>
> Dear Reviewer xVhp,
>
> We deeply appreciate your recognition of our work and your insightful feedback, which has been essential in enhancing both the robustness and quality of our study.
>
> Thank you once again for your valuable support! We are truly thankful for your time and thoughtful consideration.
>
> Best regards,
>
>  The Authors

---

### Official Review · Reviewer_QY19 · 2024-11-04

**Soundness:** 3
**Presentation:** 2
**Contribution:** 4
**Rating:** 6
**Confidence:** 4

**Summary:**

The paper conducts a study on confirmation bias from initial responses in different multi-agent LLM setups and comes up with a technique to prevent this bias (and thus improve benchmark performance) by changing the framing of questions. It then presents very initial work on how multi-agent system performance scales with the number of agents and tokens.

**Strengths:**

- The paper presents a very comprehensive review of existing multi-agent LLM research and fits in prejudice and fidelity quite well in these settings. This provides useful context to understand the paper’s key contributions.

- The problem reframing method is reasonably novel and the experimental evaluations are comprehensive enough to demonstrate improvements with this method.

- It presents initial interesting results around differences in scaling the number of agents/LLM calls versus the number of tokens per generation. This could allow for a lot more future work in multi-agent LLM research.

- APF-Bench encompasses other benchmarks and can act as a useful starting point for similar research directions.

**Weaknesses:**

- The paper explores only problem reframing as a bias mitigation strategy. However, not every problem can be converted into a binary problem, and other strategies are not explored at all.

- The paper does not perform evaluations on any open source models.

- The refinement strategy for datasets could introduce selection bias and skew results. I would be interested in seeing results across a random subset of the test set on the benchmarks used.

- The paper spends its first 5.5 pages providing a background on the problem and multi-agent LLM settings. This takes away from its key contributions, which are limited to the problem reframing strategy and very introductory work on scaling laws around fidelity. Section 6.2 is extremely limited and does not back up its claims with linked experiments.

- The appendix presents examples of model outputs, however it does not provide examples of inputs to the models (especially in the problem reframing setting). I’ve posed questions around these examples in Questions section of my review.

**Questions:**

- Page 18, Case 2, GSM8k: Could the authors provide complete inputs to the models and their outputs for each iteration?

- Is there a hypothesis around why the results hold and such biases occur in language models? Are there reasonable tests that can be conducted around this?

- Could there exist better reframing techniques? Why was the binary reframing technique selected? Will it work for all tasks?

Update - these questions have been answered by the authors.

---

> ### Author Response · Authors · 2024-11-23
>
> > W1: The paper explores only problem reframing as a bias mitigation strategy. However, not every problem can be converted into a binary problem, and other strategies are not explored at all.
>
> **Comment**: Replied in the general response.
>
> ---
>
>
> > W2: The paper does not perform evaluations on any open-source models.
>
> **Comment**:  Thank you for your valuable feedback. We acknowledge the importance of evaluating open-source models to increase the reproducibility and accessibility of our findings. While the current study primarily utilizes proprietary models like GPT-4o and Gemini due to their advanced capabilities and relevance to state-of-the-art D2C systems, we recognize the potential benefits of including open-source models in future work. In subsequent iterations of this research, we plan to incorporate evaluations of open-source models such as LLaMA and Falcon, particularly to ensure broader applicability and transparency of our approach.
>
> ---
>
> > W3: The refinement strategy for datasets could introduce selection bias and skew results. I would be interested in seeing results across a random subset of the test set on the benchmarks used.
>
> **Comment**:  We appreciate this insightful observation. Following [1], the refinement strategy was designed to improve the focus and relevance of the dataset by prioritizing samples that were incorrectly answered in our specific tasks, as detailed in `Algorithm 1` of the paper. Notably, for the Chess Move Validity dataset, we considered all 1,000 problems (or samples) for all the experiments conducted in the paper. The reason for downsizing the GSM8K and PIQA datasets is that their performance is largely saturated for the considered LLMs. To demonstrate the efficacy of our approach, we downsampled these datasets using `Algorithm 1`. As for the StrategyQA dataset, it contains 2290 questions, making it prohibitively expensive to conduct experiments on all samples [1]. However, we believe that testing on the full dataset would help assess the robustness of our approach and verify whether the observed results consistently hold across diverse data splits. Moreover, we have conducted additional experiments on the GSM8K dataset, using all samples within the Multi-Agent Debate framework, as noted below:
>
> | **Dataset**       |   **Open Debate** |  Controlled Debate (Right) |  Controlled Debate (Wrong) |
> |-------------------|--------|--------|--------|
> | ***GSM8k (300 samples chosen using algorithm 1)*** | 89.67% | 93.00% | 93.33%|
> | ***GSM8k (all 1319 samples)*** | 93.67% | 94.67% | 95.00%|
>
> ---
>
> > W4: The paper spends its first 5.5 pages providing a background on the problem and multi-agent LLM settings. This takes away from its key contributions, which are limited to the problem reframing strategy and very introductory work on scaling laws around fidelity. Section 6.2 is extremely limited and does not back up its claims with linked experiments.
>
> **Comment**:  We appreciate the reviewer's concerns regarding the balance between background content and key contributions. The detailed background section was included to contextualize our work for a broader audience, but we recognize that it may detract from the primary contributions. In a revised version, we will streamline the background content to focus on essential context and allocate more space to elaborate on our contributions.
>
> Regarding Section 6.2, we confirm that all our claims are derived from `Figures 5 and 6` of the paper. We acknowledge this and sincerely apologize for not including the corresponding figure references to support the observations and claims related to scaling laws and fidelity. In the revised manuscript, we will address this by adding the appropriate figure references and providing quantitative evidence in the appendix, like https://openreview.net/forum?id=EP6n8LCEK6&noteId=etZ2jBIeZX.
>
> ---
>
> > W5: The appendix presents examples of model outputs; however, it does not provide examples of inputs to the models (especially in the problem reframing setting). I’ve posed questions around these examples in the Questions section of my review.
>
> **Comment**:
> We understand that providing inputs alongside outputs is crucial for a comprehensive understanding of the examples, particularly in the problem reframing setting. In future iterations, we will ensure that the appendix includes complete input-output pairs for all presented examples. This will provide greater clarity and transparency, addressing the questions raised and allowing for a more thorough evaluation of the methods proposed.
>
> ---
>
> Ref:
> [1] Yilun Du, Shuang Li, Antonio Torralba, Joshua B Tenenbaum, and Igor Mordatch. Improving factuality and reasoning in language models through multiagent debate. arXiv preprint arXiv:2305.14325,

---

> ### Author Response · Authors · 2024-11-23
>
> > Q1: Page 18, Case 2, GSM8k: Could the authors provide complete inputs to the models and their outputs for each iteration?
>
> **Comment**:
> Thank you for highlighting this. We will include complete inputs and outputs for each iteration in the GSM8k case study in the appendix of a revised version of the paper. This will ensure that readers can fully understand the iterative process and its impact on the results.
>
> ---
>
> > Q2: Is there a hypothesis around why the results hold and such biases occur in language models? Are there reasonable tests that can be conducted around this?
>
> **Comment**:
> We hypothesize that the observed biases stem from the inherent structure of pretraining data and the optimization objectives used in training language models. These factors can lead to over-representation or under-representation of certain patterns. To validate this hypothesis, we plan to conduct controlled experiments that isolate specific biases and evaluate their persistence across diverse datasets and tasks. For instance, ablation studies and interventions in pretraining data distribution could provide insights into the underlying causes of such biases.
>
> ---
>
> > Q3: Could there exist better reframing techniques? Why was the binary reframing technique selected? Will it work for all tasks?
>
> **Comment**:
> Our controlled (right or wrong) initialization through problem reframing was chosen for its simplicity and ease of implementation, making it a suitable starting point for exploring problem reframing. However, we acknowledge that more nuanced reframing techniques could yield better results, especially for complex tasks. Future work will investigate alternative reframing strategies, such as multi-dimensional reframing or task-specific dynamic reframing. We will also evaluate the generalizability of these techniques across different tasks to determine their broader applicability.

---

> ### Author Response · Authors · 2024-11-23
> **Additional Results for W4**
>
> ---
>
> `Figures 5 and 6` of the paper illustrate the averaged performance versus resource usage of various LLMs in multi-label and society-level frameworks across four datasets. These figures evaluate different parameters, including (1) the number of agents, (2) the number of debate rounds, (3) the number of tokens, and (4) LLM API calls. In the revised manuscript, we will provide quantitative measures for all the subplots in these two figures, similar to the sample table shown for `Figure 5 (b)`.
>
> ### Table R1 for `Figure 5 (b)`
> The averaged accuracy (Acc) of GPT-4o in multi-agent frameworks on four datasets, with ratios of samples where the number of rounds happend ($n$) equals and exceeds 1.
>
> For brevity:
> - **Open**: Open-ended.
> - **CR**: Controlled (right).
> - **CW**: Controlled (wrong).
>
> | **Dataset**       |       | **Open**   |        |       | **CR**     |        |       | **CW**     |        |
> |-------------------|-------|------------|--------|-------|------------|--------|-------|------------|--------|
> |                   | Acc.  | n = 1      | n > 1  | Acc.  | n = 1      | n > 1  | Acc.  | n = 1      | n > 1  |
> | ***GSM8k***       | 93.67 | 98.67%     | 1.33%  | 94.67 | 98.33%     | 1.67%  | 95.00 | 96.67%     | 3.33%  |
> | ***PIQA***        | 92.33 | 96.00%     | 4.00%  | 92.00 | 97.67%     | 2.33%  | 91.00 | 99.00%     | 1.00%  |
> | ***StrategyQA***  | 80.33 | 92.67%     | 7.33%  | 79.33 | 90.67%     | 9.33%  | 79.67 | 88.67%     | 11.33% |
> | ***Chess***       | 67.00 | 59.33%     | 40.67% | 74.67 | 65.67%     | 34.33% | 79.67 | 66.00%     | 34.00% |

---

> > ### Comment · Reviewer_QY19 · 2024-11-25
> >
> > I thank the authors for their response and have raised my score.

---

> > > ### Author Response · Authors · 2024-11-27
> > > **Thank You to Reviewer QY19**
> > >
> > > Dear Reviewer QY19,
> > >
> > > We deeply appreciate your recognition of our work and your insightful feedback, which has been essential in enhancing both the robustness and quality of our study.
> > >
> > > Thank you once again for your valuable support! We are truly thankful for your time and thoughtful consideration.
> > >
> > > Best regards,
> > >
> > > The Authors

---

### Official Review · Reviewer_EH5M · 2024-11-09

**Soundness:** 3
**Presentation:** 3
**Contribution:** 2
**Rating:** 5
**Confidence:** 4

**Summary:**

The authors investigate reasoning pathologies of a certain class of multi-agent LLM systems, Diverge-to-Converge (D2C) frameworks, both at the model- and society-level. The authors identify an inherent confirmation bias in D2C systems but results in social biases and task underperformance which can be alleviated if open-ended questions are re-phrased as binary. The authors then study the scaling laws of D2C frameworks, finding that more agents does only result in performance improvements if the system is not yet saturated but can otherwise even degrade performance. The authors suggest remedies for both these pathologies and release APF-Bench to specifically evaluate these weaknesses.

**Strengths:**

* Very timely and can provoke thought - e.g. trade-off bias/compute
* Robust evaluation across multiple datasets
* the idea to use reframing to tackle biases seems novel

**Weaknesses:**

* Could have discussed a greater variety of biases other than confirmation bias
* It isn't clear how questions of real-world importance that are open-ended can always be brought into binary form.
* conceptual advances are limited - scaling laws / reframing techniques themselves feel rather incremental

line 216 "menifest"

**Questions:**

* How do you prevent bias in the debate judgements?

---

> ### Author Response · Authors · 2024-11-23
>
> > W1: Could have discussed a greater variety of biases other than confirmation bias
>
> **Comment**:
> In `Section 2` `(lines 115–123)`, we discuss various biases examined in prior works, along with our findings regarding different biases in D2C frameworks. Specifically, we uncover an inherent confirmation bias in D2C systems and propose a problem reframing strategy to mitigate it.
>
> Additionally, in `Section 6.1.2` `(lines 444–469)`, we highlight:
> 1. **Affirmative vs. negative agent bias**: Explored in both open and controlled debate scenarios (refer to `Figure 3`).
> 2. **Social biases**: Including gender (or sex) bias, such as male or female bias (refer to `Figure 4`, right bars), and political bias, such as left- or right-wing bias (refer to `Figure 4`, middle bars).
> We believe these discussions provide a broader perspective on biases beyond confirmation bias and will clarify them further in the revised manuscript.
> ---
>
> > W2: It isn't clear how questions of real-world importance that are open-ended can always be brought into binary form.
>
> **Comment**:  Replied in the general response.
>
> ---
>
> > W3: Conceptual advances are limited—scaling laws and reframing techniques themselves feel rather incremental
>
> **Comment**:
> We acknowledge the reviewer’s concern about the perceived incremental nature of the scaling laws and reframing techniques. While these methods build upon existing frameworks, our contribution lies in their novel application to D2C systems. Specifically, we demonstrate:
> - **Scalability**: The potential and limitations of collaborative agent frameworks for real-world tasks versus simpler tasks. `[Section 6.2]`
> - **Reframing strategies**: The ability to mitigate confirmation bias while maintaining performance. `[Section 6.1.1, Lines: 419-442]`
>
> These findings provide actionable insights for improving agent collaboration and addressing biases, advancing the understanding of D2C frameworks in significant ways. We will clarify this contribution in the revised manuscript.
>
> ---
>
> > W4: Line 216 "menifest"
>
> **Comment**:
> Thank you for pointing out the typographical error. The spelling of "menifest" has been corrected to "manifest" in the revised manuscript. `[Line 216]`
>
> ---
>
> > Q1: How do you prevent bias in the debate judgments?
>
> **Comment**:
> We appreciate the reviewer’s interest in preventing bias in debate judgments. Our current work focuses on uncovering and addressing confirmation bias through a problem reframing strategy. While this strategy effectively mitigates confirmation bias in D2C systems, we acknowledge the need for broader investigations into bias prevention in debate judgments.
>
> Future studies will explore additional mechanisms for detecting and addressing subtle biases, and we plan to address these challenges in subsequent work.

---

> > ### Comment · Reviewer_EH5M · 2024-11-26
> >
> > I thank the authors for their response. While I acknowledge the authors' clarifications, overall I believe the paper's contributions are nevertheless on the incremental side. A paper that I would feel comfortable with accepting would need to demonstrate some kind of dynamic adaptation to the authors' findings, including using fine-tuning or other post-training approaches to improve the weaknesses of D2C frameworks uncovered. However, in line with ICLR guidelines, I believe that such extensions are out of scope of the current submission. I keep my score.

---

> > > ### Author Response · Authors · 2024-11-27
> > >
> > > Thank you for your thoughtful feedback and the opportunity to discuss the contributions of our work further,  particularly in the context of modern multi-agent systems (MAS) and Large Language Models (LLMs).
> > >
> > > 1. **Main Scientific Contribution**: The core contribution of our study is identifying a significant confirmation bias inherent in the widely adopted Diverge-to-Converge (D2C) frameworks within MAS. Our analysis reveals that the encouragement of diverse thinking among agents, a common feature of these frameworks, paradoxically leads to confirmation bias. This insight is crucial as it highlights a fundamental issue in a growing field.
> > >
> > > 2. **Proposed Solution and Its Impact**: To address this bias, our paper presents a simple yet universally effective solution. By implementing a controlled initialization of the tasked question, we reframe the problem in a way that not only enhances the performance of D2C frameworks but also can help to mitigate broader social biases perpetuated by LLMs, such as those related to gender or ideology. This approach leads to better outcomes by aligning the divergent thinking of agents towards a more balanced convergence.
> > >
> > > 3. **Value of the Findings**: Although our research does not introduce new training techniques or LLM architectures, the significance of our findings lies in their generality and practical applicability. By demonstrating how a strategic intervention in problem framing can influence systemic biases, our work contributes a valuable perspective to the field of LLM agent research. This contribution is particularly pertinent as it provides a novel lens through which the community can reassess and refine the operational dynamics of MAS frameworks.
> > >
> > > 4. **Comparison with other LLM Debias Direction**: Thanks for pointing out the point. The studied bias is identified and tackled at the level of agent system, instead of individual LLM. We acknowledge the importance of examining biases at the data and model levels in LLM research. Our work complements these efforts by identifying and addressing biases in MAS frameworks, particularly in how agent interactions can propagate or mitigate biases. We believe that our contributions provide valuable insights that are **orthogonal to, yet supportive of**, the broader goals of reducing bias in AI systems. In the appendix, we will add a section clarifying that our findings are orthogonal to other research examining LLM biases from data or model perspectives. This section will state:
> > >
> > > ``*We acknowledge the importance of examining biases at the data and model levels in LLM research. Our work complements these efforts by identifying and addressing biases in MAS frameworks, particularly in how agent interactions can propagate or mitigate biases. We believe that our contributions provide valuable insights that are orthogonal to, yet supportive of, the broader goals of reducing bias in AI systems.*''

---

### Official Review · Reviewer_z9FH · 2024-11-09

**Soundness:** 3
**Presentation:** 3
**Contribution:** 2
**Rating:** 5
**Confidence:** 3

**Summary:**

This paper examines the limitations of Diverge-to-Converge (D2C) frameworks in large language model (LLM) agents, focusing on prejudice and fidelity. It reveals a confirmation bias in D2C systems that hampers performance and amplifies social biases, but reframing open-ended problems as binary questions mitigates these effects. The study also shows that increasing the number of agents only improves performance under unsaturated conditions. Additionally, the authors introduce APF-Bench, a benchmark to evaluate these weaknesses, providing insights for building better collaborative AI systems.

**Strengths:**

1. It uncovers and addresses *confirmation bias* in D2C frameworks, providing practical solutions to mitigate performance issues and social biases.
2. The study examines both*prejudice* and *fidelity*, offering a detailed understanding of D2C frameworks across multiple levels.
3. By demonstrating how reframing problems into binary questions improves fairness and effectiveness, the research has real-world applicability.
4. The development of *APF-Bench* as a dedicated tool for evaluating D2C systems is a valuable resource for future research.
5. The analysis of scaling laws provides essential guidelines for optimizing agent collaboration in different task scenarios.

**Weaknesses:**

1. Limited Real-World Testing: The findings might lack generalizability if not tested in diverse, real-world multi-agent scenarios.
2. Potential Oversimplification: Reframing problems as binary questions may oversimplify complex tasks, possibly limiting the depth of solutions.
3. Scalability Constraints: The performance degradation observed in saturated systems indicates a limitation in scaling D2C frameworks effectively.
4. Bias Mitigation Trade-offs: While the approach reduces biases, it may inadvertently introduce new limitations or biases in certain contexts.

**Questions:**

* Why do you study D2C frameworks rather than other MAS frameworks? Is D2C a typical and widely adopted MAS framework? What are the incentives behind this choice?
* What is the main *contribution in scientificity* that the paper claims? This paper does a lot of evaluation and analysis on different LLMs, but they are the existing ones. Could you provide insights into designing LLMs that can inherently avoid or mitigate confirmation bias? Or, can you give a discussion on the underlying causes of such bias, which could possibly arise at the data level or the pre-training/fine-tuning level instead of solely empirical discovery?
* See also Weaknesses for other questions.

---

> ### Author Response · Authors · 2024-11-23
>
> > W1: Limited Real-World Testing.
>
> Comment: Replied in the general response.
>
> ---
>
> > W2: Potential Oversimplification
>
> Comment: Replied in the general response.
>
>
> > W3: Scalability Constraints
>
> **Observation**: The performance degradation observed in saturated systems indicates a limitation in scaling D2C frameworks effectively.
>
> **Comment**:
> Our experimental observations reveal that scaling improves model performance in more complex tasks, such as Chess Move validity. Adding more agents significantly enhances strategic diversity in these scenarios. However, saturation occurs in simpler tasks; for instance, with the PIQA dataset, performance saturates when adding more than four agents.
>
> Thus, while scalability constraints may appear in simpler tasks, they are less prominent for complex, real-world problems, where scaling continues to contribute to performance gains.
>
> ---
>
> > W4: Bias Mitigation Trade-offs
>
> **Observation**: While the approach reduces biases, it may inadvertently introduce new limitations or biases in certain contexts.
>
> **Comment**:
> We appreciate the opportunity to address the potential trade-offs involved in bias mitigation. While our framework demonstrates efficacy in reducing biases, we acknowledge the possibility of introducing new limitations or biases, particularly in under-represented domains or tasks.
>
> In the revised manuscript, we will:
> - Discuss how task-specific characteristics may influence the redistribution or amplification of biases.
> - Highlight potential limitations in monitoring and mitigating emerging biases during scaling or deployment.
>
> These additions will provide a more balanced perspective on the advantages and trade-offs of our approach.
>
> ---
>
> > Q1: Why do you study D2C frameworks rather than other MAS frameworks? Is D2C a typical and widely adopted MAS framework? What are the incentives behind this choice?
>
> **Comment**:
> Thank you for your question regarding our selection of the Diverge-to-Converge (D2C) framework for our study.
> 1. **Adoption and Typicality of D2C**: D2C is indeed a typical and widely adopted MAS framework. We conceptualize this adoption as part of a broader trend where more MAS frameworks are embracing the D2C paradigm. This trend is exemplified by the frameworks we have explored in our paper, including self-consistency, consultancy, debate, and LLM agents society, which all follow the D2C approach. Some later follow-up works include [1-4].
>
> 2. **Benefits and Observations of D2C**: Based on these D2C frameworks, we have been able to identify inherent characteristics such as prejudice and fidelity, which arise from the framework’s encouragement of agent divergence. This divergence allows for a broad exploration of solutions, which is crucial for potential improvement.
>
> [1] Wang, Junlin, et al. "Mixture-of-Agents Enhances Large Language Model Capabilities." arXiv preprint arXiv:2406.04692 (2024).
>
> [2] Li, Dawei, et al. "SMoA: Improving Multi-agent Large Language Models with Sparse Mixture-of-Agents." arXiv preprint arXiv:2411.03284 (2024).
>
> [3] Li, Yunxuan, et al. "Improving Multi-Agent Debate with Sparse Communication Topology." arXiv preprint arXiv:2406.11776 (2024).
>
> [4] Zhang, Guibin, et al. "Cut the crap: An economical communication pipeline for llm-based multi-agent systems." arXiv preprint arXiv:2410.02506 (2024).
>
> ---

---

> > ### Author Response · Authors · 2024-11-27
> >
> > > Q2: What is the main contribution in scientificity that the paper claims? This paper does a lot of evaluation and analysis on different LLMs, but they are the existing ones. Could you provide insights into designing LLMs that can inherently avoid or mitigate confirmation bias? Or, can you give a discussion on the underlying causes of such bias, which could possibly arise at the data level or the pre-training/fine-tuning level instead of solely empirical discovery?
> >
> > **Comment**:
> > Thank you for the opportunity to clarify the primary scientific contributions of our paper, particularly in the context of modern multi-agent systems (MAS) and Large Language Models (LLMs).
> > 1. **Main Scientific Contribution**: The core contribution of our study is identifying a significant confirmation bias inherent in the widely adopted Diverge-to-Converge (D2C) frameworks within MAS. Our analysis reveals that the encouragement of diverse thinking among agents, a common feature of these frameworks, paradoxically leads to confirmation bias. This insight is crucial as it highlights a fundamental issue in a growing field.
> >
> > 2. **Proposed Solution and Its Impact**: To address this bias, our paper presents a simple yet universally effective solution. By implementing a controlled initialization of the tasked question, we reframe the problem in a way that not only enhances the performance of D2C frameworks but also can help to mitigate broader social biases perpetuated by LLMs, such as those related to gender or ideology. This approach leads to better outcomes by aligning the divergent thinking of agents towards a more balanced convergence.
> >
> > 3. **Value of the Findings**: Although our research does not introduce new training techniques or LLM architectures, the significance of our findings lies in their generality and practical applicability. By demonstrating how a strategic intervention in problem framing can influence systemic biases, our work contributes a valuable perspective to the field of LLM agent research. This contribution is particularly pertinent as it provides a novel lens through which the community can reassess and refine the operational dynamics of MAS frameworks.
> >
> > 4. **Comparison with other LLM Debias Direction**: Thanks for pointing out the point. The studied bias is identified and tackled at the level of agent system, instead of individual LLM. We acknowledge the importance of examining biases at the data and model levels in LLM research. Our work complements these efforts by identifying and addressing biases in MAS frameworks, particularly in how agent interactions can propagate or mitigate biases. We believe that our contributions provide valuable insights that are **orthogonal to, yet supportive of**, the broader goals of reducing bias in AI systems. In the appendix, we will add a section clarifying that our findings are orthogonal to other research examining LLM biases from data or model perspectives. This section will state:
> >
> > ``*We acknowledge the importance of examining biases at the data and model levels in LLM research. Our work complements these efforts by identifying and addressing biases in MAS frameworks, particularly in how agent interactions can propagate or mitigate biases. We believe that our contributions provide valuable insights that are orthogonal to, yet supportive of, the broader goals of reducing bias in AI systems.*''

---

> > > ### Author Response · Authors · 2024-11-27
> > >
> > > Dear Reviewer z9FH,
> > >
> > > I hope this message finds you well. We have addressed the concerns raised in our revised manuscript and would greatly appreciate your review and further comments.
> > >
> > > Thank you for your time and expertise.
> > >
> > > Best regards,
> > >
> > > The authors

---

> > > > ### Author Response · Authors · 2024-12-02
> > > > **Feeback on rebuttal**
> > > >
> > > > Dear reviewer,
> > > >
> > > > Thank you once again for your time in reviewing our paper and providing valuable feedback. As the discussion period ends tomorrow, we are reaching out to see if you have any further questions or pending issues.
> > > >
> > > > We have aimed to address your comments regarding the oversimplification of problems, scalability limitations, and the trade-offs involved in bias mitigation. The paper has also been revised to incorporate these considerations.
> > > >
> > > > Please let us know if you have any follow-up comments or require additional clarifications.
> > > >
> > > > Best regards,
> > > >
> > > > The authors

---

### Author Response · Authors · 2024-11-24
**General Response**

### Common Questions

- **`W1 (z9FH):`** Limited Real-World Testing: The findings might lack generalizability if not tested in diverse, real-world multi-agent scenarios.
- **`W2 (z9FH):`** Potential Oversimplification: Reframing problems as binary questions may oversimplify complex tasks, possibly limiting the depth of solutions.
- **`W2 (EH5M):`** It isn't clear how questions of real-world importance that are open-ended can always be brought into binary form.
- **`W1 (QY19):`** The paper explores only problem reframing as a bias mitigation strategy. However, not every problem can be converted into a binary problem, and other strategies are not explored at all.
- **`W2 (xVhp):`** In more complex scenarios, problem reframing is difficult.

---

**Comment**:

Apologies for any confusion regarding the oversimplification of reframing problems as binary questions. We acknowledge that not all tasks can be reduced to binary questions such as “yes/no” or “true/false.” Instead, we present **controlled initialization** through problem reframing for multi-agent systems by leveraging the **potential solution space** of a given problem.

### Dataset Examples
In this study, we consider four diverse datasets (for confirmation bias) that address unique, real-world task types of importance [line 266]:

1. **PIQA** `[Lines: 270–273]`:
   Contains questions with a **binary solution space**, where either "statement-1" or "statement-2" is correct.

2. **StrategyQA** `[Lines: 274–277]`:
   Also contains questions with a **binary solution space**, where the answer is "Yes" or "No."

3. **GSM8K** `[Lines: 278–281]`:
   Includes questions with a **non-binary solution space**; the solution lies in $ \mathbb{R} $ (real numbers).

4. **Chess Move Validity** `[Lines: 282–285]`:
   Similar to GSM8K, this dataset has a **non-binary solution space**. While it is commonly stated that there are 64 possible answers formatted as $[a-h][1-8]$, representing potential chess moves, the actual number of valid moves may vary depending on the specific state of the chessboard (e.g., piece positions, legal moves). Therefore, the solution space dynamically adjusts to the context of the game. Each generated answer was deemed correct as long as it was one of the valid answers in the sequence.


### Controlled Initialization Framework
Let $Q$ denote a question, $C_A$ the correct answer space, and $C_W$ the wrong answer space. Controlled initialization is structured as follows:
- For **controlled wrong initialization**, the prompt is:
  _“Is $C_W$ the correct answer to the question?”_
- For **controlled right initialization**, the prompt is:
  _“Is $C_A$ the correct answer to the question?”_

For binary tasks like **PIQA** and **StrategyQA**, where $C_W = \sim C_A$, the solution space is straightforward. However, for non-binary tasks:
- **GSM8K**: $C_A \subset \mathbb{R}$ (correct numerical solution), and $C_W = \mathbb{R} \setminus C_A$.
- **Chess Move Validity**: $C_A$ is the set of valid answers out of 64 possible moves, and $C_W$ encompasses the complement of valid moves.

### Broader Applicability and Frameworks
These controlled initializations are initiated on the **affirmative side** during the starting round for both:
1. **Multi-label frameworks** (e.g., Self-consistency, Debate, Consultancy).
2. **Society-label frameworks** (e.g., CAMEL, LangChain, AutoGen).

By explicitly tailoring the initialization to the solution space, we maintain flexibility to address task-specific complexity. In the revised manuscript, we have incorporated these clarifications and further detailed the methodologies discussed above to emphasize the adaptability of our approach.

---

### Manuscript Updates
We have marked out the updated contents in blue in the pdf. Specifically, the updates include:

1. More detailed dataset description on the searching space of solutions, in Section 5.

2. Detailed explanation on the controlled initialization, in Appendix A.

3. Further discussion on contribution, with the comparison with other LLM Debias Directions, in Appendix E.

4. Numerical results for figures, in Appendix F.

5. More detailed revisions including typos, layout adjustment, revised figure captions, etc.

---

### Meta-Review · Area_Chair_o5zJ · 2024-12-19

**Metareview:**

While this work tackles an important and interesting topic in so-called multi-agent LLM systems, I believe the current work is not ready to be published. The current work somewhat overstates its contributions by framing the paper as a broad study of prejudice and bias in multi-agent systems (as evident in the title), but only looks at confirmation bias (as pointed out by Reviewer EH5M). The experiments in the paper primarily make speculative claims about the underlying reason for the observed bias, based on the final task performance, while foregoing the opportunity to look deeper into the inference chains leading to these results. A quantitative analysis of the intermediate outputs of these multi-agent systems would provide a richer understanding of the bias studied in this work.

**Additional Comments On Reviewer Discussion:**

Most reviewers point out that this work overly simplifies the problem initially motivated in the paper. In particular I am aligned with Reviewer EH5M's concerns, which were not sufficiently addressed in the authors' rebuttal.

---

### Decision · Program_Chairs · 2025-01-22

Reject